# The intragenus and interspecies quorum-sensing autoinducers exert distinct control over *Vibrio cholerae* biofilm formation and dispersal

**Andrew A. Bridges**[1,2], **Bonnie L. Bassler**[1,2]*

**1** Department of Molecular Biology, Princeton University, Princeton, New Jersey, United States of America,
**2** Howard Hughes Medical Institute, Chevy Chase, Maryland, United States of America

* bbassler@princeton.edu

**Data Availability Statement:** All relevant data are within the manuscript and its Supporting Information files.

## Abstract

*Vibrio cholerae* possesses multiple quorum-sensing (QS) systems that control virulence and biofilm formation among other traits. At low cell densities, when QS autoinducers are absent, *V. cholerae* forms biofilms. At high cell densities, when autoinducers have accumulated, biofilm formation is repressed, and dispersal occurs. Here, we focus on the roles of two well-characterized QS autoinducers that function in parallel. One autoinducer, called *cholerae* autoinducer-1 (CAI-1), is used to measure *Vibrio* abundance, and the other autoinducer, called autoinducer-2 (AI-2), is widely produced by different bacterial species and presumed to enable *V. cholerae* to assess the total bacterial cell density of the vicinal community. The two *V. cholerae* autoinducers funnel information into a shared signal relay pathway. This feature of the QS system architecture has made it difficult to understand how specific information can be extracted from each autoinducer, how the autoinducers might drive distinct output behaviors, and, in turn, how the bacteria use QS to distinguish kin from nonkin in bacterial communities. We develop a live-cell biofilm formation and dispersal assay that allows examination of the individual and combined roles of the two autoinducers in controlling *V. cholerae* behavior. We show that the QS system works as a coincidence detector in which both autoinducers must be present simultaneously for repression of biofilm formation to occur. Within that context, the CAI-1 QS pathway is activated when only a few *V. cholerae* cells are present, whereas the AI-2 pathway is activated only at much higher cell density. The consequence of this asymmetry is that exogenous sources of AI-2, but not CAI-1, contribute to satisfying the coincidence detector to repress biofilm formation and promote dispersal. We propose that *V. cholerae* uses CAI-1 to verify that some of its kin are present before committing to the high-cell–density QS mode, but it is, in fact, the broadly made autoinducer AI-2 that sets the pace of the *V. cholerae* QS program. This first report of unique roles for the different *V. cholerae* autoinducers suggests that detection of kin fosters a distinct outcome from detection of nonkin.

**Funding:** This work was supported by the Howard Hughes Medical Institute, NIH Grant 5R37GM065859, National Science Foundation Grant MCB-1713731, and a Max Planck-Alexander von Humboldt research award to BLB. AAB is a Howard Hughes Medical Institute Fellow of the Damon Runyon Cancer Research Foundation, DRG-2302-17. The content is solely the responsibility of the authors and does not necessarily represent the official views of the National Institutes of Health. The funders had no role in study design, data collection and analysis, decision to publish, or preparation of the manuscript.

**Competing interests:** The authors have declared that no competing interests exist.

**Abbreviations:** AI-2, autoinducer-2; a.u., arbitrary unit; CAI-1, *cholerae* autoinducer-1; DPD, 4,5-dihydroxy-2,3-pentanedione; DPO, 3,5-dimethylpyrazin-2-ol; HCD, high cell density; IO, Instant Ocean; LB, lysogeny broth; LCD, low cell density; mNG, mNeonGreen; OD, optical density; PMSF, phenylmethylsulfonyl fluoride; QS, quorum sensing; RLU, relative light unit; SOE, splicing overlap extension; sRNA, small regulatory RNA; Tdh, threonine dehydrogenase; WT, wild type.

## Introduction

Bacteria communicate and orchestrate collective behaviors using a process called quorum sensing (QS). QS relies on the production, release, and group-wide detection of extracellular signaling molecules called autoinducers. QS allows bacteria to assess the cell density and the species composition in the local environment and change their behavior accordingly [1,2]. Frequently, QS controls the development of biofilms, which are surface-associated communities of bacteria that secrete an adhesive extracellular matrix [3,4]. Biofilms are beneficial in many contexts; for example, microbiota of the digestive tract exist in biofilms, but biofilms can also be harmful, for example, in infections [5]. Cells in biofilms display striking differences from their planktonic counterparts, including extracellular matrix production and a dramatic tolerance to environmental perturbations, including antibiotic treatment [4,6]. Despite the extraordinary importance of bacterial biofilms, we know only a few key facts about their development: matrix production is required, and QS-mediated communication can be involved in regulating biofilm formation and dispersal [4,7,8].

The pathogen and model QS bacterium *V. cholerae* forms biofilms in all of its niches [4,9]. *V. cholerae* strains locked in the low-cell–density (LCD) QS mode avidly form biofilms, while strains locked in the high-cell–density (HCD) QS mode are incapable of forming biofilms [3]. While these findings show an overarching role for QS in repressing biofilm formation at HCD, they are incomplete because they were obtained from *V. cholerae* mutants locked in the LCD or HCD QS mode that are thus unable to progress through the normal QS program. Furthermore, how *V. cholerae* cells disperse from biofilms and the role played by QS in dispersal have only recently begun to be addressed [10]. Here, we establish a simple microscopy-based assay with wild-type (WT) *V. cholerae* that allows us to examine the full biofilm lifecycle and assess the role of QS in both biofilm formation and biofilm dispersal.

The canonical *V. cholerae* QS pathway is composed of two well-characterized autoinducer–receptor pairs that function in parallel to funnel cell-density information internally to control gene expression (Fig 1A) [11]. One autoinducer–receptor pair consists of *cholerae* autoinducer-1 (CAI-1; (*S*)-3-hydroxytridecan-4-one), produced by the synthase, CqsA, and detected by the two-component sensor-histidine kinase, CqsS [12,13]. CAI-1 is an intragenus signal for *Vibrios*. The second autoinducer–receptor pair is comprised of autoinducer-2 (AI-2; (2*S*,4*S*)-2-methyl-2,3,3,4-tetrahydroxytetrahydrofuran borate), produced by the broadly conserved synthase, LuxS, and detected by LuxPQ [11,14,15]. LuxP is a periplasmic binding protein that interacts with AI-2. LuxP ligand occupancy is monitored by LuxQ, a transmembrane two-component sensor-histidine kinase [16,17]. AI-2 is produced by diverse bacterial species and is considered to be a QS autoinducer that conveys interspecies information [15]. Two other receptors, CqsR and VpsS, have recently been shown to feed information into this network; however, the identities of their cognate autoinducers are not known (Fig 1B) [18,19]. Ethanolamine functions as a surrogate agonist for CqsR [20]. All four receptors act as kinases at LCD in their unliganded states [21–23]. They funnel phosphate through the phosphotransfer protein LuxU to the response regulator LuxO, which, via a set of small regulatory RNAs (sRNAs) called the Qrr sRNAs, drives the production of the LCD master regulator AphA and represses production of the HCD master regulator HapR [24–27]. Under these conditions, behaviors including biofilm formation and virulence factor production are undertaken (Fig 1A, left) [3,11]. When bound to their cognate autoinducers, the receptors act as phosphatases [22]. LuxO is dephosphorylated, AphA production is terminated, and HapR production is activated [27]. In this situation, HCD behaviors are enacted, and, germane to this work, virulence and biofilm formation are repressed and *V. cholerae* disperses from existing biofilms (Fig 1A, right) [10,28]. It has long been puzzling why the autoinducer signals are processed via the

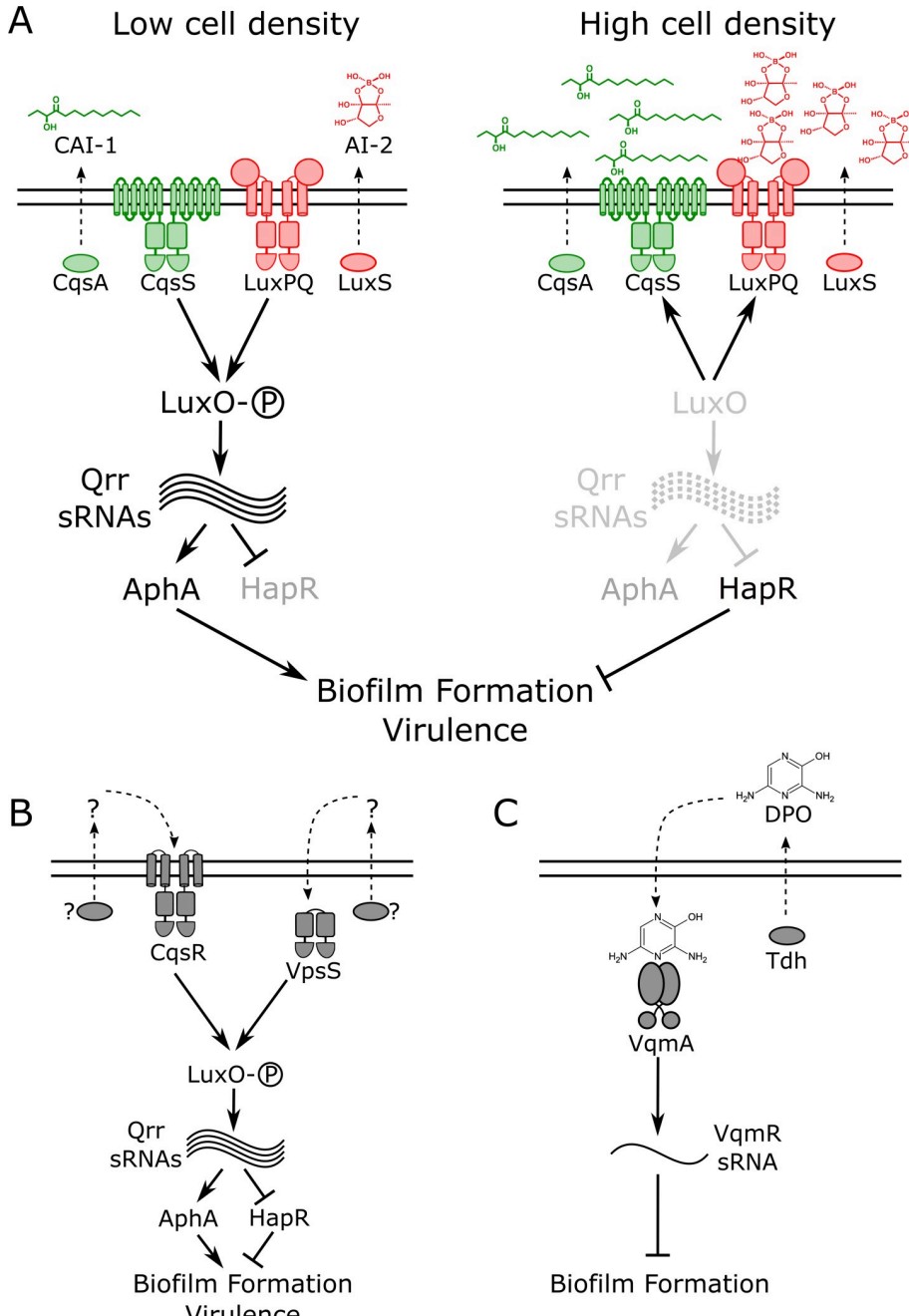

**Fig 1. Simplified *V. cholerae* QS circuits.** (A) Two established autoinducer–receptor pairs control QS behaviors in *V. cholerae*. One autoinducer–receptor pair consists of CAI-1, which is synthesized by CqsA and detected by the two-component sensor-histidine kinase, CqsS. The second autoinducer–receptor pair is AI-2, produced by LuxS and detected by LuxPQ, also a two-component sensor-histidine kinase. At LCD (left), both receptors act as kinases that promote phosphorylation of the response regulator, LuxO. Phosphorylated LuxO activates expression of genes encoding regulatory RNAs called the Qrr sRNAs. The Qrr sRNAs activate production of the LCD master regulator, AphA, and repress production of the HCD master regulator, HapR. These conditions drive biofilm formation and virulence factor production. At HCD (right), the autoinducer-bound receptors act as phosphatases that strip phosphate from LuxO, resulting in AphA repression and HapR production, conditions that promote the free-swimming, planktonic lifestyle and repression of virulence. (B) Two additional QS receptors, VpsS and CqsR, also funnel information into LuxO. Their cognate autoinducers and autoinducer syntheses are not known. (C) A recently discovered QS pathway consists of the autoinducer DPO, synthesized by Tdh, and its partner receptor VqmA. At HCD, DPO-bound VqmA activates expression of a gene encoding a sRNA called VqmR. VqmR represses biofilm formation. AI-2, autoinducer-2; CAI-1, *cholerae* autoinducer-1; DPO, 3,5-dimethylpyrazin-2-ol; HCD, high cell density; LCD, low cell density; QS, quorum sensing; sRNA, small regulatory RNA; Tdh, threonine dehydrogenase.

identical, shared pathway in *V. cholerae* because this system architecture is not obviously conducive to gleaning specific information from each autoinducer.

Recently, we discovered another *V. cholerae* QS pathway that functions independently of the above QS system (Fig 1C) [29–31]. In this case, the autoinducer, called DPO (3,5-dimethyl-pyrazin-2-ol), binds to a cytoplasmic transcriptional regulator called VqmA. The VqmA–DPO complex activates expression of a gene encoding a regulatory RNA called VqmR. VqmR represses genes required for biofilm formation. Thus, the DPO–VqmA–VqmR circuit also represses biofilm formation at HCD.

Here, we develop a real-time assay to measure WT *V. cholerae* biofilm formation and dispersal. The assay does not demand the use of locked QS mutants, allowing us to examine the role QS plays over the entire biofilm lifecycle. We find that the CAI-1 and AI-2 QS pathways control biofilm formation, while the DPO pathway has no effect in this assay. The AI-2 receptor LuxPQ strongly promotes biofilm formation at LCD when the ligand is absent, while the CAI-1 receptor, CqsS, is incapable of driving biofilm formation at LCD. The mechanism underlying the effect stems from markedly different cell-density thresholds required for autoinducer detection by the two QS receptors, with the kin CAI-1 autoinducer threshold being achieved at much lower cell densities than that of the nonkin AI-2 autoinducer. Nonetheless, we show that both autoinducers must be present simultaneously for repression of biofilm formation to occur, suggesting that the QS system functions as a coincidence detector. Collectively, our results show that a small number of kin must be present to activate *V. cholerae* QS, but the pace at which QS occurs is driven by the timing by which the interspecies AI-2 autoinducer accumulates. To our knowledge, this is the first report of unique roles for the different *V. cholerae* autoinducers, and our findings imply that detection of kin fosters a different outcome than detection of nonkin.

## Results

### A new biofilm growth and dispersal assay for WT *V. cholerae*

In *V. cholerae* biofilm studies to date, researchers have overwhelmingly employed either hyper-biofilm–forming *V. cholerae* strains that are locked at LCD and incapable of QS and biofilm dispersal, or they have used fluid flow to wash autoinducers away from growing WT biofilms, in effect locking the *V. cholerae* cells at LCD [10,32–35]. While these strategies have accelerated studies of early *V. cholerae* biofilm formation and enabled identification and characterization of biofilm matrix components, QS, which is known to control the process, has not been systematically examined in a WT *V. cholerae* strain capable of naturally transitioning between LCD and HCD behavior as biofilms form and disperse. We developed simple static growth conditions that permitted WT *V. cholerae* biofilm formation and dispersal. Our strategy allows endogenously produced autoinducers to accumulate and drive changes in QS-controlled gene expression in living, growing WT *V. cholerae* biofilms; this is, to our knowledge, a first for the field. We used *V. cholerae*, O1 biovar El Tor strain C6706, which is known to transition between the biofilm and free-swimming states. This strain, when inoculated at LCD onto glass coverslips in minimal medium, grew into discrete biofilms, and subsequently, biofilm dispersal occurred (Fig 2A and S1 Movie). Many biofilms were simultaneously imaged over time using low-magnification brightfield microscopy. With these images, we could measure bulk biofilm biomass accumulation by performing intensity-based segmentation of the biofilms coupled with quantitation of the attenuation of light that occurred due to biofilm growth. This procedure revealed that WT biofilms grew to peak biomass at an average of approximately 8–9 h after inoculation, and complete dispersal occurred by 13 h (Fig 2B). To confirm that the imaged cell clusters were indeed biofilms, we conducted identical

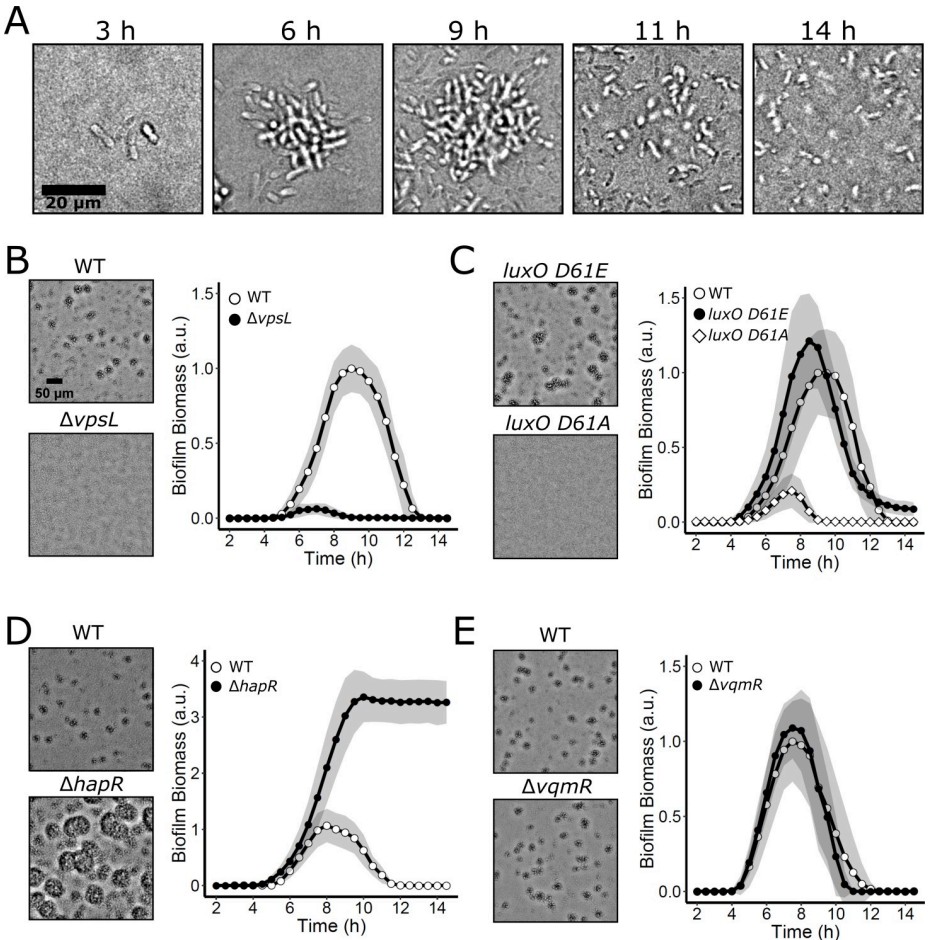

**Fig 2. *V. cholerae* biofilm formation and dispersal under static growth conditions.** (A) Time course of a representative WT *V. cholerae* biofilm lifecycle as imaged by brightfield microscopy using high magnification (63× objective). (B) Left panels: brightfield projections of *V. cholerae* biofilms in the indicated strains after 9 h of growth at 30°C, imaged using low-magnification (10× objective). Right panel: quantitation of *V. cholerae* WT and Δ*vpsL* biofilm biomass over time. (C) As in B for *V. cholerae* WT and QS mutants locked in LCD (*luxO D61E*) and HCD (*luxO D61A*) modes. (D) As in B for *V. cholerae* WT and the LCD-locked Δ*hapR* strain. (E) As in B for *V. cholerae* WT and the Δ*vqmR* strain. Data are represented as means normalized to the peak biofilm biomass of the WT strain in each experiment. In all cases, *n* = 3 biological and *n* = 3 technical replicates, ± SD (shaded). Numerical data are available in S1 Data. a.u., arbitrary unit; HCD, high cell density; LCD, low cell density; QS, quorum sensing; WT, wild type.

experiments using a Δ*vpsL* mutant strain that is incapable of producing the major polysaccharide component of the extracellular matrix required for biofilm formation [32]. No biofilm formation was detected in this mutant (Fig 2B and S1 Movie). To validate the method, we show that complementation of the Δ*vpsL* mutant via expression of *vpsL* from an ectopic chromosomal locus restored biofilm formation (S1A Fig).

To assess how this biofilm growth and dispersal assay compares to previous methods used for measuring QS control of *V. cholerae* biofilm formation, we analyzed the biofilm formation and dispersal phenotypes of mutant strains locked in the QS LCD and HCD modes. As mentioned, QS represses biofilm formation at HCD, and consistent with this pattern, both the LCD-locked *luxO D61E* mutant carrying a LuxO-P mimetic and the Δ*hapR* mutant lacking the HCD master QS regulator (see Fig 1A) accumulated greater biofilm biomass than WT *V. cholerae*. Moreover, neither mutant fully dispersed (Fig 2C and 2D and S1 Movie). Notably,

the phenotype of the Δ*hapR* strain was more extreme in its preference for the biofilm state than that of the *luxO D61E* strain, consistent with the downstream position and direct function of HapR in regulation of biofilm formation. Specifically, LuxO D61E drives constitutive production of the Qrr sRNAs (Fig 1A) [36]. The Qrr sRNAs activate translation of AphA and repress translation of HapR, and they positively and negatively regulate other targets [37,38]. Thus, in the *luxO D61E* strain, unlike in the Δ*hapR* strain, some HapR is present that can activate biofilm dispersal, and, moreover, other Qrr-regulated targets also promote biofilm dispersal in the LuxO D61E mutant. These features of the QS circuit have been reported previously and underlie the difference in phenotypes between the two mutants [37]. Importantly, complementation of the Δ*hapR* mutant by ectopic expression restored near-WT timing of biofilm dispersal (S1B Fig). A strain carrying the nonphosphorylatable *luxO D61A* allele, which is locked in the HCD QS mode, failed to form appreciable biofilms (Fig 2C and S1 Movie). Together, these data verify that the *V. cholerae* canonical QS system shown in Fig 1A controls biofilm formation in our assay. Below, we probe the roles of the individual QS circuits. To assess the contribution from the DPO–VqmA–VqmR pathway, we measured biofilm biomass over time in a strain lacking the VqmR regulatory RNA. At HCD, the Δ*vqmR* mutant cannot repress biofilm genes (Fig 1C). The *vqmR* mutant displayed WT biofilm formation and dispersal behaviors (Fig 2E), suggesting that the DPO–VqmA–VqmR pathway does not influence biofilm phenotypes under our assay conditions and/or when the canonical QS system is present. We do not study the DPO–VqmA–VqmR pathway further in the present work.

## AphA and HapR exhibit inverse production patterns during biofilm development, and AphA predominates in biofilms

The functioning of the canonical QS system is well established in WT *V. cholerae* cells under planktonic growth conditions: AphA is highest in abundance at LCD, and its levels decline as cell density increases. Conversely, HapR is present at low levels at LCD, and it accumulates with increasing cell density [27,39]. We wondered whether this inverse relationship also exists in growing WT biofilms. To examine the patterns of the two regulators, we measured the abundances of AphA and HapR during biofilm formation by building strains carrying either chromosomal *aphA-mNG* (mNeonGreen) or chromosomal *hapR-mNG* at their native loci. We also introduced a red fluorescent reporter, *mRuby3*, driven by the constitutive promoter, $P_{TAC}$, into each strain for normalization. We reasoned that the relative amounts of the AphA-mNG and mRuby3 or HapR-mNG and mRuby3 in cells could be used as a proxy for QS state. Using the above low-magnification imaging technique, coupled with confocal fluorescence microscopy and single-biofilm segmentation, we measured the fluorescence outputs from the reporters in individual biofilms over time (Fig 3). The AphA-mNG signal increased following initiation of the biofilm assay, an increase that occurred prior to the start of image acquisition, and subsequently declined 4-fold over the lifetime of the biofilm relative to the constitutive reporter (Fig 3A and 3B). Conversely, following dilution of the HCD overnight culture into the biofilm assay, the HapR-mNG output decayed during early biofilm formation. Thereafter, HapR-mNG was only minimally produced until about 5–6 h of biofilm development. At that time, the HapR-mNG fluorescence signal began to increase, and it peaked immediately prior to dispersal (Fig 3C and 3D). Subsequently, HapR-mNG was abundantly present in cells that had become planktonic, while AphA-mNG was undetectable in planktonic cells (S2 Movie). The ratio of AphA-mNG/HapR-mNG throughout the time course revealed that for the first 3.5 h of biofilm development, there was 10- to 17-fold more AphA than HapR (Fig 3D inset). The ratio then steadily declined, and immediately preceding dispersal, the AphA/HapR ratio was approximately 1:1. We conclude that the majority of the *V. cholerae* biofilm lifetime is

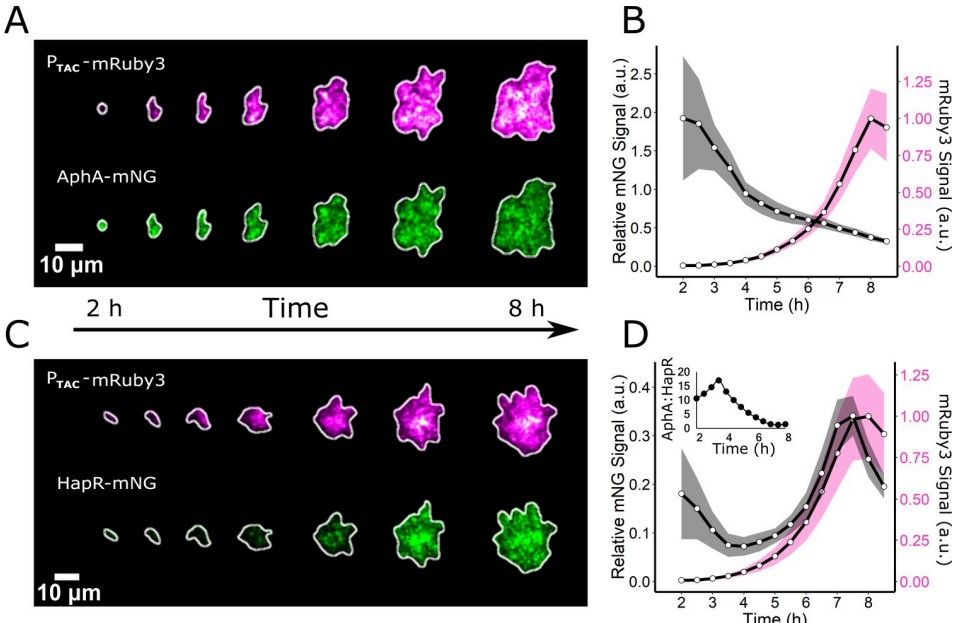

**Fig 3. AphA and HapR abundances vary inversely during biofilm formation.** (A) Representative image series showing the formation of an individual biofilm harboring the constitutive reporter $P_{TAC}$-mRuby3 and AphA-mNG. (B) Quantitation of the AphA-mNG fluorescence (black line) relative to the control mRuby3 fluorescence (magenta line) over the course of biofilm development. *n* = 24 biofilms from three biological replicates. (C and D) As in A and B, respectively, for HapR-mNG. Inset in (D) represents the AphA-mNG/HapR-mNG ratio over time. Shading in B and D represents SD. Numerical data are available in S1 Data. a.u., arbitrary unit; mNG, mNeonGreen.

spent in an AphA-dominated regime. Only immediately preceding dispersal does the level of HapR increase, resulting in the transition to the planktonic lifestyle. These results suggest that AphA and HapR levels vary inversely in response to changes in cell density and that relationship is maintained in both biofilm and planktonic cells. Thus, the core behavior of the QS system is conserved in both growth modes.

## AI-2 represses WT *V. cholerae* biofilm formation

We wondered how autoinducers influence *V. cholerae* biofilm formation and dispersal. As mentioned, two QS receptors, VpsS and CqsR, have recently been discovered that feed information into the canonical QS pathway, but their cognate autoinducers and autoinducer synthases are not identified (Fig 1B) [18]. For that reason, we cannot control autoinducer production for these two circuits, nor can we quantify their inputs into QS-driven biofilm behavior. To avoid confounding issues arising from signaling by two unidentified autoinducers, in some experiments, we deleted the *vpsS* and *cqsR* genes so that inputs from the two unknown cues were eliminated, allowing us to quantitatively assess the activities of CAI-1 and AI-2. In every experiment, we specify whether the *vpsS* and *cqsR* genes are present or not.

To probe the individual roles of CAI-1 and AI-2 in repression of biofilm formation and driving biofilm dispersal, we built reporter strains that exclusively respond to only one of these two autoinducers. Each reporter strain possesses a single QS receptor, but it lacks the corresponding autoinducer synthase. Thus, only exogenously supplied autoinducer can activate QS and only via the single remaining receptor. To our surprise, addition of synthetic CAI-1 at a saturating concentration of 5 μM ($EC_{50}$ = 32 nM, [40]) at the initiation of biofilm formation had no effect on biofilm development or dispersal in the CAI-1 reporter strain (Δ*vpsS*, Δ*cqsR*,

ΔluxQ, ΔcqsA) because the results were identical to when solvent was added (S2A Fig). In contrast, administration of 5 μM of a structurally unrelated CqsS agonist ($EC_{50}$ = 9 nM [40]), 1-ethyl-*N*-{[4-(propan-2-yl)phenyl]methyl}-1*H*-tetrazol-5-amine (which we call Mimic$^{CAI-1}$ for simplicity) markedly reduced biofilm formation (S2A Fig). We confirmed that our synthetic CAI-1 is fully active in this reporter strain by monitoring bioluminescence emission from a chromosomally integrated luciferase (*luxCDABE*) operon driven by the QS-controlled native promoter. This reporter is routinely used as a heterologous readout for HapR-controlled QS activity in *V. cholerae* [28]. When grown in shaken, planktonic conditions, both CAI-1 and Mimic$^{CAI-1}$ induced an approximately 1,000-fold increase in light production by the CAI-1 reporter strain, although Mimic$^{CAI-1}$ activated the reporter earlier, at a lower cell density (S2B and S2C Fig, respectively). These results suggest that exogenously supplied synthetic CAI-1 is only inactive under biofilm growth conditions. We suspect that differences in the physical properties of synthetic CAI-1 and Mimic$^{CAI-1}$ are responsible for this discrepancy, and we address these differences in the Discussion. In experiments requiring autoinducer supplementation, we supply exogenous Mimic$^{CAI-1}$ in place of CAI-1 to activate signaling transduction through CqsS for the remainder of this work. Addition of saturating AI-2 (5 μM; $EC_{50}$ = 21 nM, as measured in *V. harveyi* [17]) to the corresponding AI-2 reporter strain (ΔvpsS, ΔcqsR, ΔcqsS, ΔluxS) dramatically reduced biofilm formation and, moreover, activated the *lux* reporter roughly 1,000-fold, showing that AI-2 is active in both assays (S3A and S3B Fig, respectively).

We next explored how exogenous provision of Mimic$^{CAI-1}$ or AI-2 influences the WT *V. cholerae* biofilm program, in the case in which all four QS receptors are present and all of the autoinducers are also endogenously produced and accumulate naturally over time. The architecture of the *V. cholerae* QS system is arranged such that all four autoinducers feed information into the same signal integrator, LuxO, and as such, the expectation is that administration of additional Mimic$^{CAI-1}$ or AI-2 should prevent biofilm formation and/or promote dispersal (Fig 1A and 1B). To the contrary, we found that the addition of 5 μM Mimic$^{CAI-1}$ to *V. cholerae* cells that naturally produce CAI-1 and AI-2 had little effect on WT biofilm biomass accumulation or dispersal (Fig 4A and 4B). However, addition of 5 μM AI-2 repressed biofilm formation and promoted premature biofilm dispersal (Fig 4A and 4B). We obtained identical results when the *vpsS* and *cqsR* genes were present and when they had been deleted, showing that input from these two circuits is negligible under these conditions (S4 Fig). To confirm that AI-2 caused its effect via the *V. cholerae* QS system, we assayed whether AI-2 could repress biofilm formation in the *V. cholerae luxO D61E* strain that is locked in the LCD QS mode and does not respond to autoinducers [36]. AI-2 had no effect on biofilm formation or dispersal in this strain (Fig 4C). Therefore, AI-2 requires a functional QS system to drive changes in *V. cholerae* biofilm behavior. These results suggest that exogenously supplied AI-2, but not Mimic$^{CAI-1}$, should foster premature induction of HapR, the downstream master regulator of the QS HCD state. Indeed, saturating AI-2 caused HapR-mNG production to increase after only 3 h of biofilm growth, and by 6 h, HapR-mNG levels were 8-fold higher than in untreated biofilms and 3-fold higher than in Mimic$^{CAI-1}$-treated biofilms (Fig 4D and 4E). To our knowledge, these findings represent the first case in which AI-2/LuxPQ activity has a stronger effect than CAI-1/CqsS activity on *V. cholerae* QS behavior.

The difference in strengths of the CAI-1 and AI-2 autoinducers on biofilm repression was unexpected. We wondered whether the dominance of the AI-2 signal was specific to the biofilm formation/dispersal process or if other *V. cholerae* QS-controlled traits were likewise differentially controlled. To explore this possibility, we focused on virulence factor production, which, like biofilm formation, is activated at LCD and repressed at HCD (Fig 1A). To monitor virulence, we introduced a 3×FLAG epitope onto the C-terminus of the major subunit of the

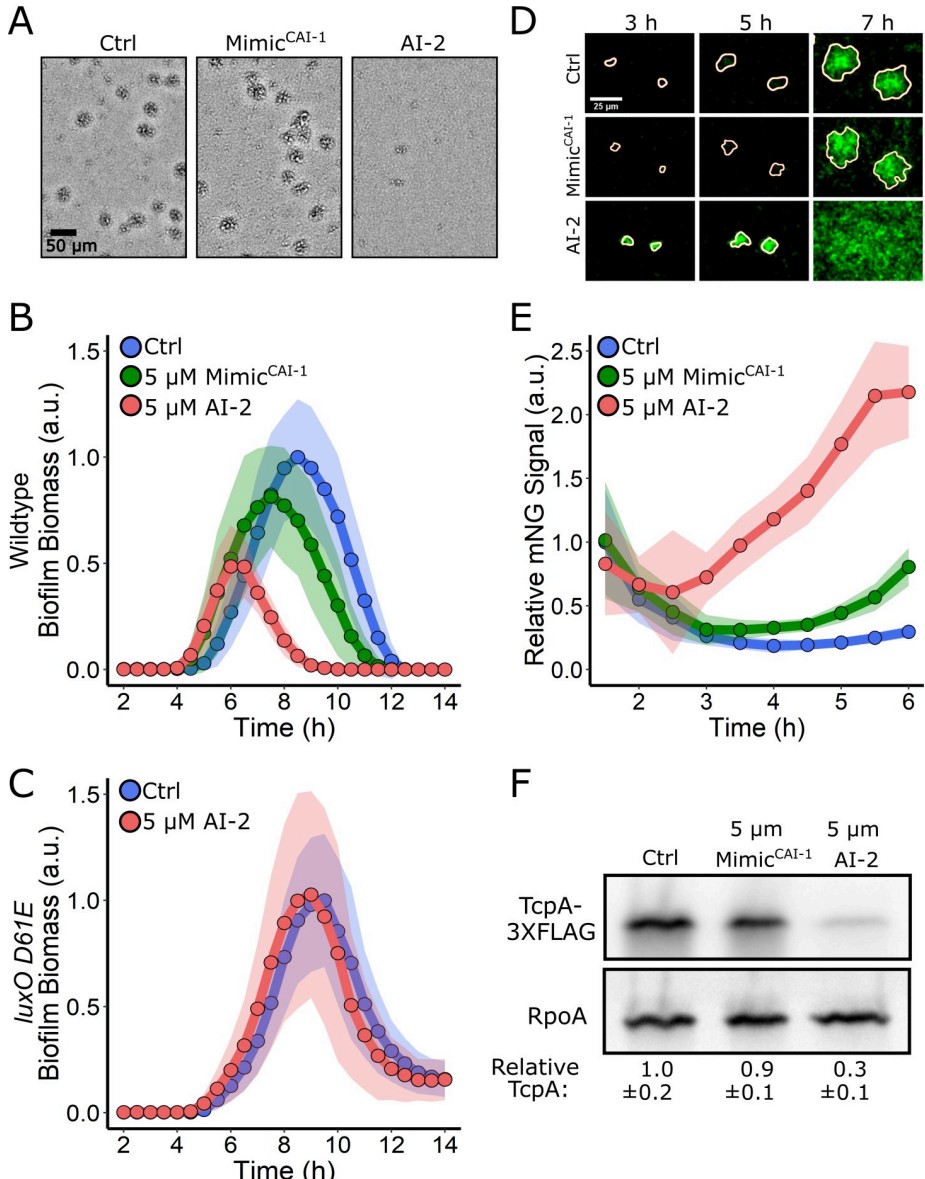

**Fig 4. Exogenous AI-2 represses WT *V. cholerae* biofilm formation, but Mimic[CAI-1] does not.** (A) Representative projections of WT *V. cholerae* treated with 0.25% DMSO (Ctrl), 5 μM Mimic[CAI-1], or 5 μM AI-2 after 9 h of biofilm growth at 30°C. (B) Quantitation of biofilm biomass for WT *V. cholerae* treated with 0.25% DMSO (Ctrl), 5 μM Mimic[CAI-1], or 5 μM AI-2 over time. Data are represented as means normalized to the peak biofilm biomass of the DMSO Ctrl. In all cases, $n = 3$ biological and $n = 3$ technical replicates, ± SD (shaded). (C) As in B for the *V. cholerae luxO D61E* strain treated with 0.25% DMSO (Ctrl) or 5 μM AI-2. (D) Representative images of WT *V. cholerae* producing HapR-mNG after treatment as in B. (E) Quantitation of HapR-mNG signal relative to the control, $P_{TAC}$-mRuby3 signal over the course of biofilm development following treatment as in B. $n = 24$ biofilms from three biological replicates. Data are normalized to the initial intensity of the sample to which DMSO was added. (F) Representative western blot for TcpA-3×FLAG in WT *V. cholerae* treated with 0.25% DMSO (Ctrl), 5 μM Mimic[CAI-1], or 5 μM AI-2. RpoA was used as the loading control. Quantification represents three biological replicates for each condition. Values were normalized to the Ctrl. Numerical data are available in S1 Data. AI-2, autoinducer-2; a.u., arbitrary unit; CAI-1 *cholerae* autoinducer-1; Ctrl, control; mNG, mNeonGreen; WT, wild type.

toxin-coregulated pilus, TcpA, and placed the *tcpA-3×FLAG* construct onto the chromosome of an otherwise WT *V. cholerae* strain. Under growth conditions conducive to production of

virulence factors, western blot analysis showed TcpA-3×FLAG was produced by the strain in the control experiment in which 0.25% DMSO solvent was added (Fig 4F). Exogenous addition of 5 μM Mimic$^{CAI-1}$ did not alter production of TcpA-3×FLAG. In contrast, treatment with 5 μM AI-2 resulted in a 70% decrease in TcpA-3×FLAG production. These results show that exogenous AI-2 is the dominant QS autoinducer controlling TcpA production, analogous to the results presented in Fig 4B for biofilm formation.

Together, the experiments in Fig 4 show that AI-2 dominates over CAI-1 under biofilm and virulence conditions. We next monitored expression of the chromosomally integrated QS-controlled *lux* reporter to assess the roles of the two autoinducers under conventional, shaken planktonic growth conditions. In contrast to biofilm formation and virulence factor production, which are repressed by HapR at HCD, autoinducer accumulation drives HapR to activate *lux* gene expression at HCD. Specifically, in bioluminescence assays, light output is high immediately following dilution of an HCD overnight planktonic culture. Thereafter, light production declines precipitously because the autoinducers have been diluted to below their levels of detection. As the cells grow, endogenously produced autoinducers accumulate, and light production again commences. Thus, a "U"-shaped curve is a hallmark QS-activated gene expression pattern (S5A Fig). To examine the effect of each autoinducer on *lux* activation, we administered 5 μM of either Mimic$^{CAI-1}$ or AI-2 to WT *V. cholerae* carrying the *lux* reporter. The results mirror those shown for biofilm formation in Fig 4B except there is activation, not repression, of behavior. Here, at LCD, addition of AI-2, but not Mimic$^{CAI-1}$, stimulated a 10-fold enhancement in light production irrespective of whether the *vpsS* and *cqsS* genes are present or not (S5B and S5C Fig, respectively). Together, our results exploring QS repression of biofilm formation, repression of virulence factor production, and activation of light production demonstrate that WT *V. cholerae* shows little response to exogenously supplied Mimic$^{CAI-1}$ but is highly sensitive to exogenously supplied AI-2.

## CqsS is activated at extremely low cell densities

Given that LuxPQ and CqsS relay information to the same response regulator, LuxO, it was not obvious how exogenous AI-2 could so dominate the WT QS phenotypes. First, regarding biofilms, one possibility is that LuxPQ kinase activity is required for biofilm formation at LCD while CqsS kinase activity is dispensable. If so, exogenous AI-2, but not Mimic$^{CAI-1}$, would drive repression of biofilm formation and activation of biofilm dispersal. To test this idea, we measured biofilm formation and dispersal in strains possessing only a single autoinducer synthase–receptor pair, either LuxS/AI-2 and LuxPQ (designated AI-2$^{S+R+}$) or CqsA/CAI-1 and CqsS (designated CAI-1$^{S+R+}$) and compared them to the strain containing both synthase–receptor pairs (designated CAI-1$^{S+R+}$, AI-2$^{S+R+}$). In all cases, the strains lacked the VpsS and CqsR receptors (see the schematic in Fig 5A for the depiction of the strains). Importantly, in these experiments, we did not supply exogenous autoinducers. The AI-2$^{S+R+}$ strain accumulated biofilm biomass and dispersed identically to the CAI-1$^{S+R+}$, AI-2$^{S+R+}$ strain (Fig 5A, middle panel). By contrast, the CAI-1$^{S+R+}$ strain was defective in the ability to form biofilms, and it dispersed prematurely (Fig 5A, middle panel). This experiment shows that the LuxPQ kinase can drive *V. cholerae* biofilm formation at LCD, while the CqsS kinase cannot. To determine if this relationship is unique to biofilm growth, or if it also applies to planktonic behaviors, we measured the ability of the same strains to activate *lux* expression. The AI-2$^{S+R+}$ strain showed the WT (i.e., CAI-1$^{S+R+}$, AI-2$^{S+R+}$) pattern for light production (Fig 5A, right panel). By contrast, at LCD, the CAI-1$^{S+R+}$ strain produced 100-fold more light than the AI-2$^{S+R+}$ and CAI-1$^{S+R+}$, AI-2$^{S+R+}$ strains, resulting in a pattern of light production nearly indistinguishable from a strain lacking the four QS

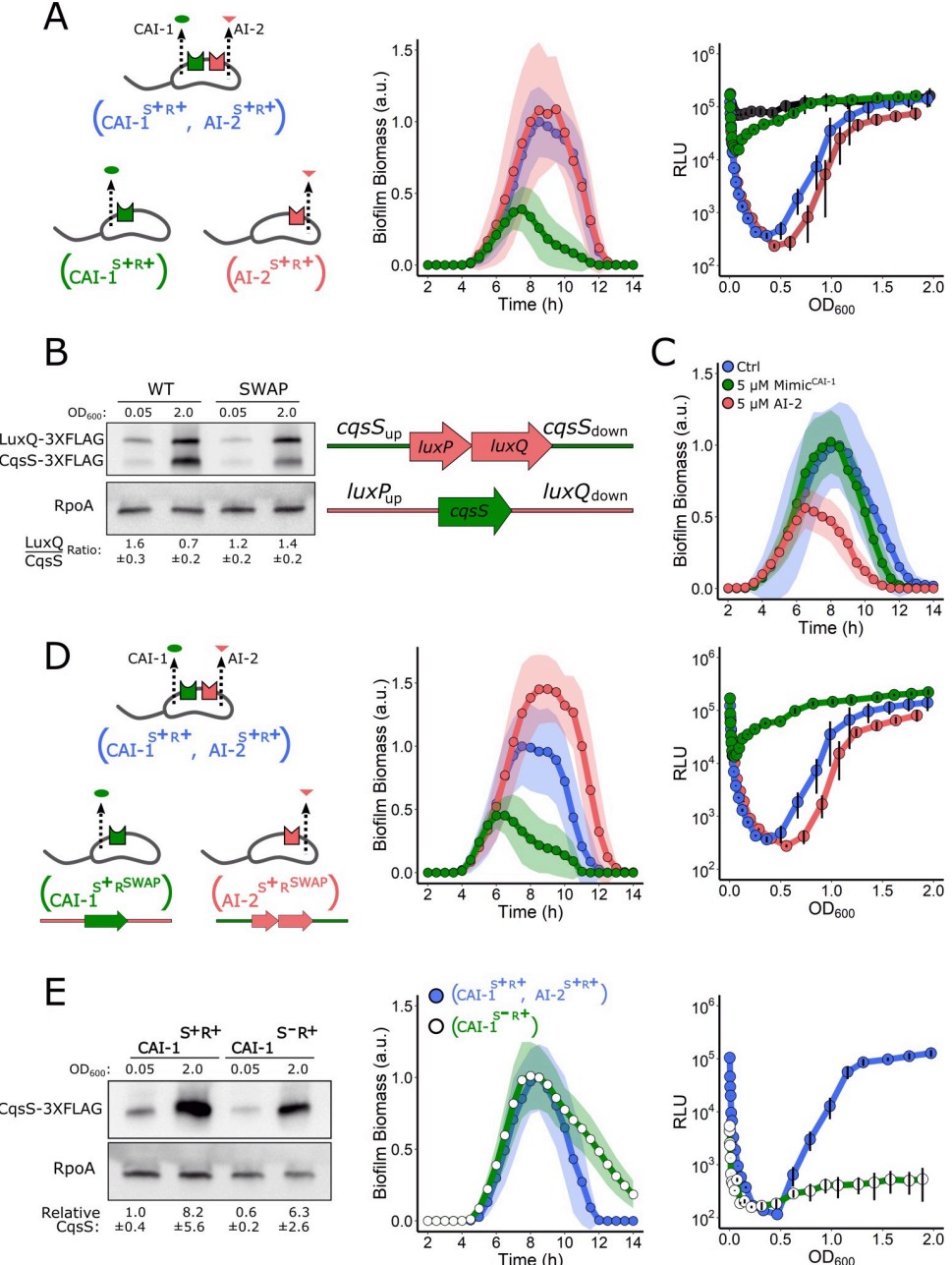

**Fig 5. LuxPQ but not CqsS drives LCD QS behaviors.** (A) Left panel: schematic representing a *V. cholerae* strain that contains both QS circuits and strains that produce and detect only a single autoinducer. Middle panel: Quantitation of biofilm biomass over time for the strain possessing both QS circuits (AI-2$^{S+R+}$, CAI-1$^{S+R+}$; blue), only the AI-2 QS circuit (AI-2$^{S+R+}$ red), and only the CAI-1 QS circuit (CAI-1$^{S+R+}$; green). Right panel: the corresponding *lux* patterns for the strains in the middle panel. The additional black curve shows the result for the *V. cholerae* strain lacking all four QS receptors (ΔvpsS, ΔcqsR, ΔluxQ, ΔcqsS). (B) Left panel: representative western blot for a strain containing CqsS-3×FLAG and LuxQ-3×FLAG produced from their native loci (WT) and for a strain in which their genomic positions had been exchanged (SWAP). RpoA was used as the loading control. Quantification of the LuxQ/CqsS ratio is based on three biological replicates for each condition. Right panel: schematic showing exchange of the *cqsS* and *luxPQ* genomic locations. (C) Quantitation of biofilm biomass for the strain with the exchanged LuxPQ and CqsS alleles (CAI-1$^{S+RSWAP}$, AI-2$^{S+RSWAP}$) treated with 0.25% DMSO (Ctrl), 5 μM Mimic$^{CAI-1}$, or 5 μM AI-2 over time. (D) Left panel: schematic representing a *V. cholerae* strain that contains both QS circuits and strains that produce and detect only a single autoinducer in which the receptor genes are expressed from the exchanged loci. Middle panel: quantification of biofilm biomass over time for a *V. cholerae* strain possessing both QS circuits (CAI-1$^{S+R+}$, AI-2$^{S+R+}$; blue), the AI-2 circuit only, with *luxPQ* expressed from the *cqsS* locus (AI-2$^{S+RSWAP}$; red), and the CAI-1 circuit only,

with *cqsS* expressed from the *luxPQ* locus (CAI-1$^{S+RSWAP}$; green). Right panel: the corresponding *lux* patterns for the strains in the middle panel. (E) Left panel: representative western blot showing CqsS-3×FLAG levels in the *V. cholerae* CAI-1$^{S+R+}$ and CAI-1$^{S-R+}$ strains. Quantification is based on three biological replicates for each condition. Middle panel: quantitation of biofilm biomass over time for the *V. cholerae* CAI-1$^{S+R+}$, AI-2$^{S+R+}$ (blue circles, blue line) and CAI-1$^{S-R+}$ (open circles, green line) strains. Right panel: The corresponding *lux* patterns for the strains in the middle panel. In all biofilm measurements, data are represented as means normalized to the peak biofilm biomass of the CAI-1$^{S+R+}$, AI-2$^{S+R+}$ strain and $n = 3$ biological and $n = 3$ technical replicates, ± SD (shaded). In all *lux* experiments, RLUs are defined as light production (a.u.) divided by OD$_{600}$ and $n = 3$ biological replicates, and error bars represent SD. Numerical data are available in S1 Data. AI-2, autoinducer-2; a.u., arbitrary unit; CAI-1, *cholerae* autoinducer-1; Ctrl, control; LCD, low cell density; OD, optical density; QS, quorum sensing; RLU, relative light unit; WT, wild type.

receptors. The mutant that has no QS receptors, CAI-1$^{S+R-}$, AI-2$^{S+R-}$, lacks all QS kinase inputs and therefore produces maximal constitutive bioluminescence (Fig 5A, right panel, depicted in black). Lastly, we assessed TcpA-3×FLAG levels by western blot as a measure of virulence factor production in each of these strains. The WT (i.e., CAI-1$^{S+R+}$, AI-2$^{S+R+}$) and AI-2$^{S+R+}$ strains produced substantial levels of TcpA-3×FLAG (S6 Fig), consistent with their ability to establish the LCD gene expression program for biofilm formation and bioluminescence emission. By contrast, the CAI-1$^{S+R+}$ strain and the strain lacking all QS receptors (CAI-1$^{S+R-}$, AI-2$^{S+R-}$) had levels of TcpA-3×FLAG that were almost undetectable. Together, these results show that LuxPQ establishes the LCD QS mode while the CqsS receptor does not do so in biofilms, for virulence, or in the planktonic cell light production assay.

One mechanism that could underlie the, respectively, strong and weak effects of LuxPQ and CqsS in control of LCD QS behaviors is that *cqsS* is not sufficiently expressed at LCD, effectively making CqsS absent and therefore unable to promote the LCD QS state. If, by contrast, LuxPQ is present at LCD, its kinase could be exclusively responsible for promoting LCD QS behaviors. Western blot analysis of a strain containing 3×FLAG-tagged LuxQ and 3×FLAG-tagged CqsS produced from their native loci revealed that LuxQ was roughly twice as abundant as CqsS at LCD, while CqsS was in excess of LuxQ at HCD (Fig 5B, left panel). We next exchanged the genomic positions of *cqsS-3×FLAG* and *luxPQ-3×FLAG*, placing each receptor gene under the other's promoter (Fig 5B, schematic). In this case, LuxQ and CqsS were present at roughly equal levels at LCD and LuxQ was in slight excess of CqsS at HCD (Fig 5B, right side of blot). Provision of exogenous AI-2 or Mimic$^{CAI-1}$ to the strain containing the exchanged alleles (CAI-1$^{S+RSWAP}$, AI-2$^{S+RSWAP}$) revealed that AI-2 remained the dominant autoinducer in LCD repression of biofilm formation (Fig 5C). Moreover, in strains carrying a single synthase–receptor pair in which the genomic locations of the receptors had been exchanged (designated CAI-1$^{S+RSWAP}$ and AI-2$^{S+RSWAP}$; see schematic in Fig 5D), little biofilm formation occurred when *cqsS* was expressed from the *luxPQ* locus, while biofilm biomass accumulated in excess of that in WT *V. cholerae* when *luxPQ* was expressed from the *cqsS* locus (Fig 5D, middle panel). Consistent with this finding, in the luciferase assay, the strains containing the singly exchanged receptors behaved the same as when the respective receptor gene was expressed from its native site (Fig 5D, right panel). These results show that the WT relative abundances of the QS receptors cannot explain the difference between the CqsS and LuxPQ kinase activities and, in turn, their influence on QS at LCD.

We considered two other possibilities to explain the variation in QS receptor kinase activity at LCD. First, either CqsS is an intrinsically poor kinase when unliganded, so it cannot drive the LCD state, or second, CqsS binds to the CAI-1 autoinducer and switches from kinase to phosphatase mode at cell densities much lower than those traditionally considered to be LCD, so its influence over the LCD QS state is rapidly abolished as the cells grow. To distinguish between these two possibilities, we deleted the CAI-1 autoinducer synthase gene, *cqsA*, from

the CAI-1$^{S+R+}$ strain, generating the CAI-1$^{S-R+}$ strain, and we examined the ability of this strain to establish the LCD behavior. Importantly, the amount of CqsS present at LCD, as measured by western blotting, was similar in the CAI-1$^{S-R+}$ strain and that of the CAI-1$^{S+R+}$ parent strain that contains *cqsA* (Fig 5E, left panel). The CAI-1$^{S-R+}$ strain was capable of driving WT levels of biofilm formation and, moreover, exhibited a delay in dispersal (Fig 5E, middle panel, and compare these results to those shown for the CAI-1$^{S+R+}$ strain in Fig 5A, middle panel). Furthermore, the CAI-1$^{S-R+}$ strain failed to activate light production in the planktonic *lux* assay irrespective of cell density (Fig 5E, right panel). These data demonstrate that when the CAI-1 autoinducer is absent, the CqsS kinase is indeed sufficiently potent to drive the LCD QS program both on surfaces and in planktonic conditions. Thus, in strains that possess both CqsA and CqsS, at LCD, there must be enough CAI-1 autoinducer present to inhibit CqsS kinase-driven biofilm formation and prevent *lux* expression.

## The *V. cholerae* QS system is a coincidence detector

Based on the above results, we suggest that, at very low cell densities, sufficient CAI-1 is present to bind the CqsS receptor and convert it from kinase to phosphatase mode. By contrast, because the critical concentration of AI-2 required to transform LuxPQ from a kinase to a phosphatase is achieved only at higher cell densities, LuxPQ remains a kinase, enabling biofilms to form and begin to mature, for virulence factor production to occur, and in the case of luciferase, *lux* is not activated. If so, during this time window, the activities of the two receptors oppose one another. We know that kinase activity is critical for establishing the LCD QS program, and since biofilms form, and light production is off at LCD, it suggests that LuxPQ kinase overrides CqsS phosphatase. Following this same logic, we hypothesize that, if kinase activity must dominate for LCD behaviors to be undertaken, it should not matter which receptor is the kinase and which receptor is the phosphatase. To test this supposition, we measured light output in a *V. cholerae* strain possessing both QS receptors, but lacking the AI-2 synthase, LuxS (CAI-1$^{S+R+}$, AI-2$^{S-R+}$). In this case, CqsS switches from kinase to phosphatase upon CAI-1 binding, and LuxPQ is a constitutive kinase. The CAI-1$^{S+R+}$, AI-2$^{S-R+}$ strain produced approximately 1,000-fold less light than the CAI-1$^{S+R+}$, AI-2$^{S+R+}$ strain that contains both autoinducer–receptor pairs (Fig 6A). We performed the reciprocal experiment using a strain lacking the CAI-1 synthase, CqsA (CAI-1$^{S-R+}$, AI-2$^{S+R+}$). In this case, CqsS is the constitutive kinase, and LuxPQ transitions from kinase to phosphatase upon AI-2 binding. This strain also exhibited 1,000-fold reduced light production at LCD relative to the CAI-1$^{S+R+}$, AI-2$^{S+R+}$ strain (Fig 6A). Consistent with these findings, these same strains, i.e., lacking one of the autoinducer synthases, displayed mild defects in biofilm dispersal relative to the strain possessing both autoinducer–receptor pairs (S7 Fig). Together, these results show that kinase activity, irrespective of which receptor provides it, overrides phosphatase activity at LCD. Moreover, it means that both autoinducers must be present simultaneously for a robust and timely transition from LCD to HCD to occur. Thus, the *V. cholerae* QS system functions as a coincidence detector for the two autoinducer inputs.

With a coincidence detection model in mind, we predicted that the addition of exogenous AI-2 should have no effect on biofilm formation and dispersal in a strain possessing both receptors but lacking the CAI-1 synthase CqsA (CAI-1$^{S-R+}$, AI-2$^{S+R+}$). In this setup, LuxPQ would function as a phosphatase upon binding to AI-2, and the lack of the *cqsA* gene would ensure that CqsS remains a kinase at all cell densities. Thus, CqsS kinase should override AI-2–bound LuxPQ phosphatase. Indeed, Fig 6B shows that AI-2 has no effect on biofilm formation/dispersal in this strain. By contrast, simultaneous administration of Mimic$^{CAI-1}$ and AI-2 to the CAI-1$^{S-R+}$, AI-2$^{S+R+}$ strain satisfies the coincidence detector

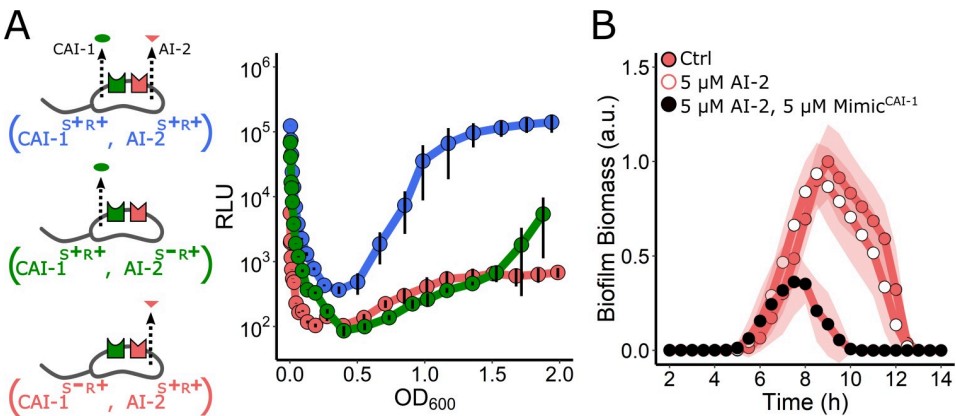

**Fig 6. The *V. cholerae* QS circuit is a coincidence detector.** (A) Left panel: schematic for strains used in the right panel, which shows the *lux* expression patterns. The strains are CAI-1$^{S+R+}$, AI-2$^{S+R+}$ (blue), CAI-1$^{S+R+}$, AI-2$^{S-R+}$ (green), and CAI-1$^{S-R+}$, AI-2$^{S+R+}$ (red). RLUs are defined as light production (a.u.) divided by OD$_{600}$. $n$ = 3 biological replicates, and error bars represent SD. (B) Quantitation of biofilm biomass for the CAI-1$^{S-R+}$, AI-2$^{S+R+}$ strain to which DMSO solvent (red circles, Ctrl), 5 µM AI-2 (white circles), or 5 µM AI-2 and 5 µM Mimic$^{CAI-1}$ (black circles) was added. Data are represented as means normalized to the peak biofilm biomass of the control, and $n$ = 3 biological and $n$ = 3 technical replicates, ± SD (shaded). AI-2, autoinducer-2; a.u., arbitrary unit; CAI-1, *cholerae* autoinducer-1; Ctrl, control; OD, optical density; QS, quorum sensing; RLU, relative light unit.

requirement, converts both CqsS and LuxPQ to phosphatase mode, and causes biofilm repression (Fig 6B). We conclude that while the *V. cholerae* QS system is a coincidence detector, the consequence of the exceedingly low cell-density activation of CqsS by endogenously produced CAI-1 makes it so that endogenous accumulation or exogenous sources of AI-2 satisfy the coincidence detector leading to HCD behaviors.

## CAI-1 activation of CqsS occurs via QS, not self-sensing

Recently, several bacterial QS circuits have been shown to be capable of self-sensing—in which an individual cell releases and detects the autoinducer that it, itself, synthesized, without sharing this autoinducer with the community—by an autocrine-like mechanism (Fig 7A) [41,42]. For self-sensing to occur, the cells must harbor sufficient levels of the receptor to capture/bind the released molecule prior to it diffusing away [43]. Importantly, self-sensing is distinct from kin sensing via QS. Kin sensing occurs when bacteria of the same or closely related species share autoinducers among the cells in the vicinity. We considered the possibility that CAI-1 could be sensed by the same cell that secretes it, potentially explaining how the CAI-1/CqsS arm of the QS system becomes activated at such low cell densities relative to the AI-2/LuxPQ circuit. On the other hand, we expected that the AI-2/LuxPQ circuit must display QS behavior rather than self-sensing, explaining why, relative to the CAI-1/CqsS circuit, the AI-2/LuxPQ arm does not engage until much higher cell densities. To explore these ideas, we examined self- versus non-self–sensing in each circuit by coculturing a "secrete-and-sense" strain (containing a single autoinducer synthase–receptor pair) with a "sense-only" strain (containing only that receptor) (Fig 7A). The rationale is that if self-sensing occurs, in coculture, the autoinducer made by the secrete-and-sense strain would trigger its HCD mode, while the sense-only strain would remain in LCD mode (Fig 7B, top). By contrast, if released autoinducer is shared between the two strains, then both the secrete-and-sense and the sense-only strains would proceed through the LCD to HCD QS program simultaneously (Fig 7B, middle). A final possibility is an intermediate state in which secrete-and-sense cells do undergo self-sensing but also share a portion of the autoinducers they make with other cells in the community.

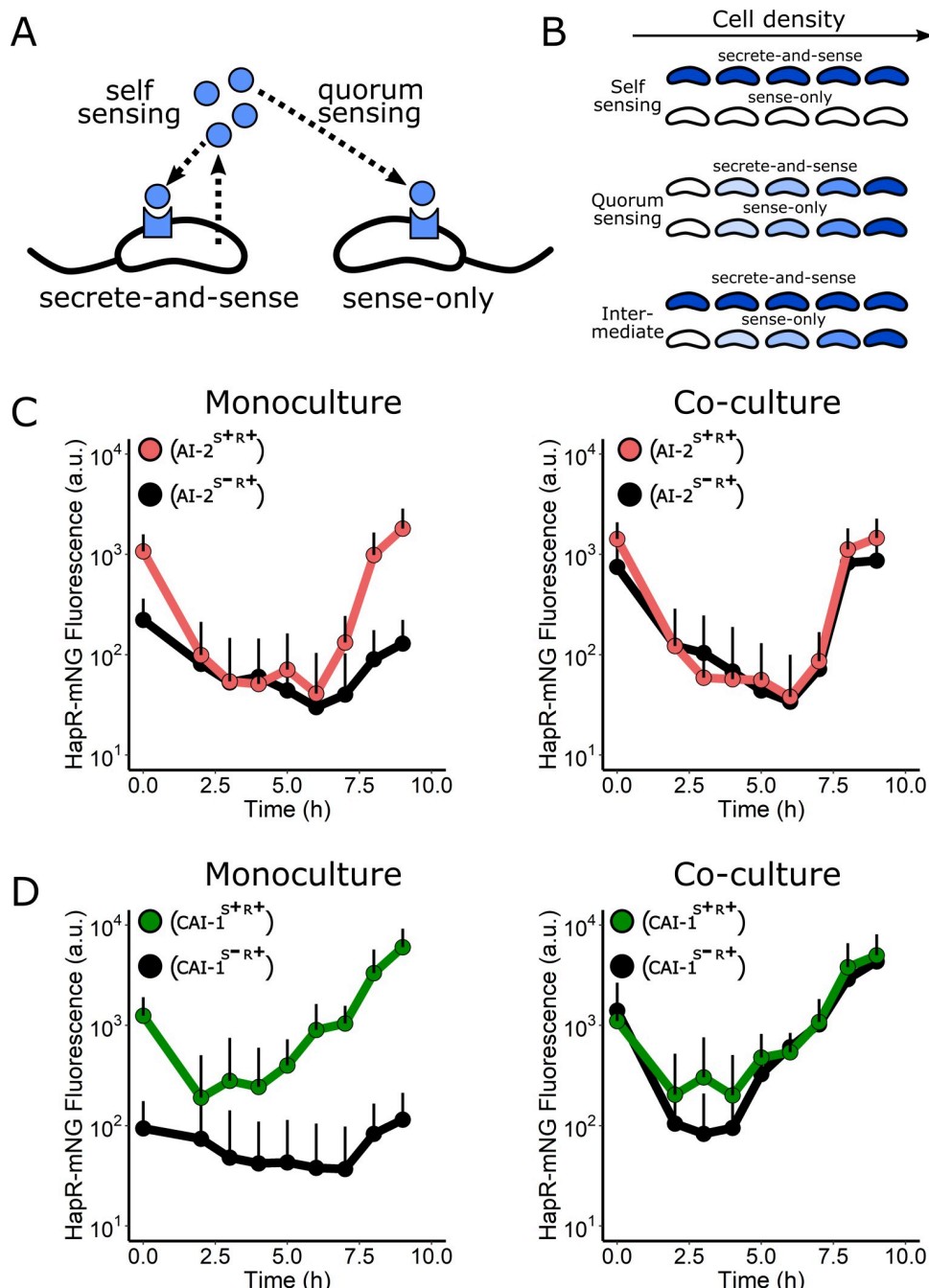

**Fig 7. The CqsS/CAI-1 circuit is primarily a QS circuit, not a self-sensing circuit.** (A) Schematic showing self-sensing and QS. See text for details. (B) Predicted HCD gene expression level (shades of blue) over increasing cell density for cocultured secrete-and-sense and sense-only strains if a circuit exhibits exclusive self-sensing behavior (top), QS behavior (middle), or an intermediate state in which both self-sensing and QS occur (bottom). (C) Left panel: average individual cell HapR-mNG fluorescence for the *V. cholerae* AI-2 $^{S+R+}$ (red) and the AI-2 $^{S-R+}$ (black) strains grown in monoculture. Right panel: the same strains grown in coculture. (D) Left panel: average individual cell HapR-mNG fluorescence for the *V. cholerae* CAI-1 $^{S+R+}$ (green) and the CAI-1 $^{S-R+}$ (black) strains grown in monoculture. Right panel: the same strains grown in coculture. Error bars represent SD of individual cell measurements at each time point. Numerical data are available in S1 Data. AI-2, autoinducer-2; a.u., arbitrary unit; CAI-1, *cholerae* autoinducer-1; mNG, mNeonGreen; QS, quorum sensing.

In this scenario, the secrete-and-sense strain would activate QS gene expression earlier than sense-only cells (Fig 7B, bottom).

We first examined self-sensing in the AI-2/LuxPQ circuit. To characterize individual cell responses following coculture, we used flow cytometry analyses to measure HapR-mNG fluorescence as a readout of HCD in the secrete-and-sense and sense-only strains. We differentiated between the strains by introducing a constitutive mRuby3 fluorescence reporter into one of the strains. As controls, we measured production of HapR-mNG in the AI-2/LuxPQ secrete-and-sense strain (AI-2$^{S+R+}$) and in the sense-only strain (AI-2$^{S-R+}$) grown in monoculture (Fig 7C). In these experiments, we diluted the cells to the very low cell density of optical density (OD)$_{600}$ = $5 \times 10^{-6}$, or roughly 2,000 cells/mL. For reference, typical *V. cholerae* QS assays are initiated at OD$_{600}$ = $5 \times 10^{-4}$, or approximately 200,000 cells/mL. Upon dilution, the secrete-and-sense AI-2$^{S+R+}$ strain repressed HapR-mNG production about 10-fold and, importantly, to the same level as the sense-only AI-2$^{S-R+}$ strain. In the AI-2$^{S+R+}$ secrete-and-sense strain, HapR-mNG production remained low for many growth cycles and began to increase only after 7 h of growth, at an approximate OD$_{600}$ of 0.1 (Fig 7C, left). In coculture, clear QS behavior occurred: HapR-mNG fluorescence in the sense-only AI-2$^{S-R+}$ strain matched that of the secrete-and-sense AI-2$^{S+R+}$ strain (Fig 7C, right). These results indicate that released AI-2 is detected equally by all cells irrespective of whether or not they can produce the autoinducer.

We next performed analogous experiments to test self-sensing versus QS by the CAI-1/CqsS circuit. We first analyzed the secrete-and-sense (CAI-1$^{S+R+}$) and sense-only (CAI-1$^{S-R+}$) strains grown in monoculture (Fig 7D, left). In this case, the secrete-and-sense CAI-1$^{S+R+}$ strain repressed HapR-mNG 5-fold by 2 h postdilution. Thereafter, HapR-mNG fluorescence rapidly increased. Notably, however, the secrete-and-sense CAI-1$^{S+R+}$ strain did not repress HapR-mNG to the level of that by the sense-only CAI-1$^{S-R+}$ strain grown alone, indicating that either a low level of self-sensing occurs or that an even greater dilution of the cells and, in turn, accumulated autoinducer is required to completely convert CqsS to the kinase mode. In coculture, HapR-mNG fluorescence followed a similar trajectory for the sense-only CAI-1$^{S-R+}$ and the secrete-and-sense CAI-1$^{S+R+}$ strains (Fig 7D, right), indicating that QS is the major driver of HapR-mNG induction in this circuit despite its early activation. We do note that the sense-only CAI-1$^{S-R+}$ strain showed modestly more repression of HapR production than the secrete-and-sense CAI-1$^{S+R+}$ strain. We interpret this result to mean that the CqsS/CAI-1 circuit does engage in a minor amount of self-sensing, likely related to the high sensitivity of this circuit. From these results, we can conclude that QS, not self-sensing, is the major signaling mechanism responsible for activation of both *V. cholerae* QS circuits; however, the two circuits are activated by their cognate autoinducers at radically different cell densities, with the CAI-1/CqsS arm being activated at much lower cell densities than the AI-2/LuxPQ circuit.

## Discussion

In this study, we present a real-time assay for WT *V. cholerae* biofilm growth and dispersal. This approach enables analysis of WT *V. cholerae* that naturally transitions from LCD to HCD and therefore progresses through the entire QS cycle. LCD-locked QS *V. cholerae* mutants were analyzed in earlier iterations of biofilm assays because their constitutive hyper-biofilm–forming phenotypes enabled imaging of biofilms as they formed. The locked LCD QS mutants were especially instructive, yielding the major matrix components and their roles, cell packing patterns, and the contributions of mechanics to biofilm morphology [4,33,34]. However, the locked LCD mutants precluded assessment of QS control over the biofilm program, and furthermore, the locked LCD mutants used in the earlier studies do not disperse from biofilms, so

the second part of the lifecycle—the transition from the biofilm to the planktonic phase—could not be accessed. Our new assay permits the study of the full biofilm program from initiation to dispersal, and moreover, mutants that are defective in particular QS components can be studied, individual cell and bulk measurements can be made, autoinducers and analogs can be supplied exogenously, and reporter genes can be monitored individually or in combination. Additionally, this assay is easily adapted to high-throughput microscopy approaches because it is performed in 96-well plates and does not require the complexities of microfluidics to deliver flow. Going forward, our intention is to use the assay with a focus on the understudied dispersal process: identifying the genes that orchestrate dispersal and the molecular mechanisms that enable cells to escape from matrix-covered sessile communities.

Using this new assay, we first confirmed that WT *V. cholerae* forms biofilms at LCD and disperses from them at HCD. We quantitatively imaged the master regulators to assess QS states in developing and dispersing biofilms. We found that the AphA-driven LCD regime spans nearly the entirety of the *V. cholerae* biofilm lifecycle. Control is passed to HapR, the HCD regulator, only immediately preceding biofilm dispersal. Investigation of the individual and collective roles of the kin (CAI-1) and nonkin (AI-2) receptors showed that they function as a coincidence detector: both autoinducers must be present simultaneously for repression of biofilms and launching of dispersal to occur.

Most surprising was our finding that, in growing biofilms, a marked asymmetry exists in QS signaling. Endogenously produced CAI-1 accumulates rapidly, activating the CqsS phosphatase early in biofilm development, whereas AI-2 does not accumulate to the threshold required to transition LuxPQ from a kinase to a phosphatase until biofilms are significantly more mature (Fig 8A). Thus, AI-2 accumulation is the limiting step for the transition from the LCD to HCD QS mode, and for driving the transition from biofilm growth to biofilm dispersal. Indeed, it is likely that the temporal offset in the accumulation of the two autoinducers is responsible for the observed asymmetry in biofilm control. Precedence for autoinducer accumulation asymmetry exists in the closely related organism, *V. harveyi*, under planktonic growth conditions [44]. Although CAI-1 inhibits the CqsS kinase when only a few thousand *V. cholerae* cells/mL are present, the CAI-1–CqsS circuit functions primarily via QS, not self-sensing, because released CAI-1 is shared between producing and nonproducing cells (Fig 7).

A longstanding mystery in the *Vibrio* QS field is how the kin (CAI-1) and nonkin (AI-2) autoinducers are decoded given that they feed information into the same regulatory network. A central question has been whether each autoinducer can uniquely modulate gene expression. The present work gives us the first clues concerning this issue. The coincidence detector property of the QS system, coupled with the dramatic difference in the cell-density–dependent activation thresholds for the two autoinducers, provides a mechanism for each autoinducer to drive unique behaviors. In so doing, each autoinducer can play a fundamentally different role in the progression from LCD to HCD QS behavior (Fig 8B). Specifically, the CAI-1/CqsS circuit has a remarkably low threshold for cell-density–dependent activation. Thus, we propose that the CAI-1/CqsS arm serves as a filter that prevents the transition to HCD mode when fewer than the critical threshold number of kin cells are present, even in scenarios in which dense populations of nonkin bacteria are present (as judged by AI-2 levels). The activity we observe for the CAI-1/CqsS circuit is consistent with theoretical work suggesting that a possible evolutionary benefit of QS is that it enables bacteria to verify the presence of related neighbors prior to committing to potentially costly group behaviors, thereby limiting benefits to "cheaters" in the community [45]. We contrast the behavior of the CAI-1/CqsS circuit to that of the AI-2/LuxPQ circuit, which has a high cell-density–dependent activation threshold. Thus, for *V. cholerae*, the buildup of AI-2 is the rate limiting step for satisfying the coincidence detector constraint. We propose that the accumulation of AI-2 sets the pace of *V. cholerae* QS.

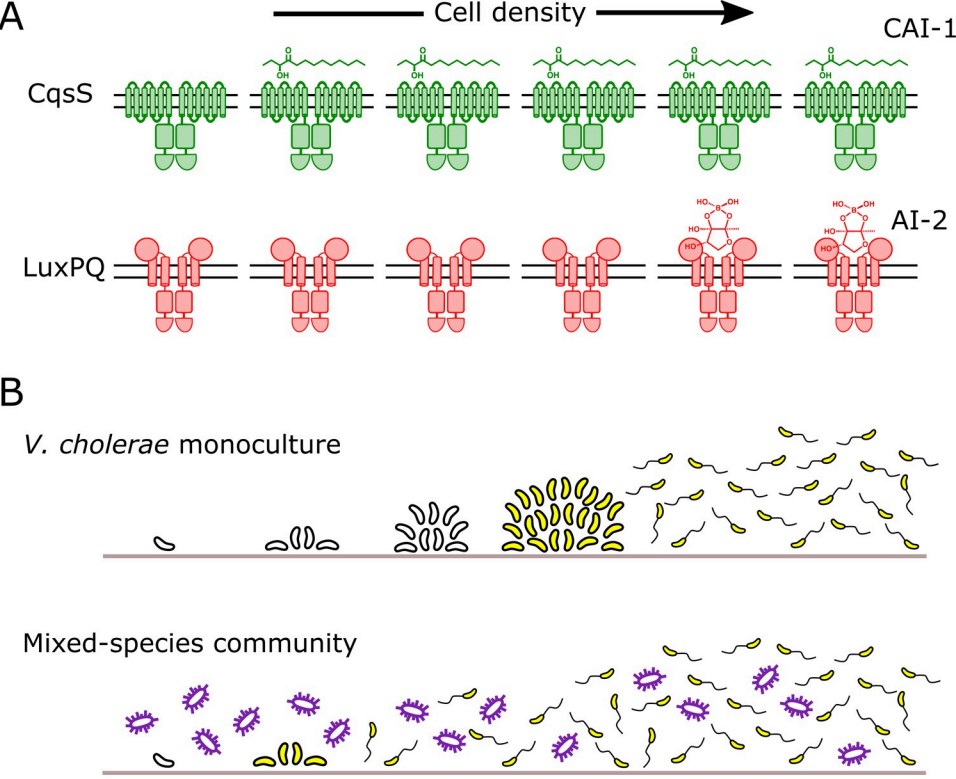

**Fig 8. Asymmetric autoinducer thresholds drive distinct intragenus and interspecies QS responses.** (A) CAI-1 produced by *V. cholerae* engages its cognate CqsS receptor at very low cell densities. In contrast, AI-2 does not accumulate to sufficient levels to engage its cognate LuxPQ receptor until much higher cell densities. (B) The consequence of asymmetric receptor occupancy coupled with the QS system functioning as a coincidence detector is that AI-2 sets the pace at which QS occurs. In *V. cholerae* monoculture (top), the absence of AI-2 at LCD is required for biofilm formation. Thus, exogenous AI-2, such as that provided in mixed-species communities by bacteria that possess LuxS, presumably represses *V. cholerae* biofilm development and/or promotes dispersal (bottom). AI-2, autoinducer-2; CAI-1, *cholerae* autoinducer-1; LCD, low cell density; QS, quorum sensing.

Our finding that endogenous production of AI-2 by *V. cholerae* does not exceed the threshold for LuxPQ activation until millions of cells/mL are present provides *V. cholerae* the capacity to tune into exogenous sources of AI-2, however, only after the requirement for the presence of CAI-1 is met. In our experiments, we supplied the AI-2 stimulus, but in natural contexts, exogenous AI-2 would be provided by other, nonkin bacteria in mixed-species communities.

Our demonstration that WT *V. cholerae* is sensitive to AI-2, but not to CAI-1, at all cell densities above a few thousand cells/mL indicates that when *V. cholerae* cell density has exceeded the CAI-1/CqsS activation threshold, the appearance of AI-2 would drive dramatic changes in gene expression (Fig 8B). We take this finding to mean that when a minority *V. cholerae* community of kin detects a majority of nonkin AI-2 producers, *V. cholerae* disperses from biofilms, exiting the current locale, presumably to identify superior territory. Indeed, our results further suggest that *V. cholerae* would only begin forming a new biofilm when it locates an unoccupied new area to colonize, as judged by the absence of autoinducers.

Intriguingly, the dominance of the LuxPQ receptor over the CqsS receptor in establishing the LCD QS program that we discover here has not been observed in a murine model of *V. cholerae* infection [18]. In infant mice, QS receptor kinase activity is required for colonization to occur. Mutant *V. cholerae* strains containing only the CqsS/CAI-1 or only the LuxPQ/AI-2 circuit can both establish infections. In the context of our current work, the finding that is particularly

surprising is that the mutant possessing only the CAI-1/CqsS circuit is capable of colonization given the propensity of the CqsS receptor to transition from kinase to phosphatase. We suspect that in this model mammalian host, perhaps CAI-1 is degraded, a host factor sequesters CAI-1, fluid flow in the gut removes CAI-1, or reduced CAI-1 production occurs. Any of these mechanisms or others would result in CqsS acting as a kinase to maintain the LCD QS state and drive biofilm formation and virulence gene expression, which are required for infection.

We were surprised that synthetic CAI-1, while active when provided to the CAI-1 reporter strain growing in the planktonic state, showed no activity when administered to growing biofilms (S2B Fig). One possibility is that the amphipathic character of CAI-1 prevents it from penetrating the biofilm matrix. A recent study suggests that endogenously produced CAI-1 partitions into outer membrane vesicles, which stabilizes the molecule and facilitates its transmission between planktonic cells [46]. Thus, it is possible that synthetic CAI-1 partitions into vesicles or makes micelles in water and becomes inaccessible to biofilm cells encapsulated in a matrix. This is clearly not the case for the polar AI-2 and Mimic$^{CAI-1}$ molecules that are active in both our planktonic and biofilms assays. While we do not know if these or other mechanisms underlie the inactivity of exogenously administered CAI-1 in our biofilm assays, using the CAI-1 surrogate Mimic$^{CAI-1}$ allowed us to overcome this experimental challenge to investigate how the activation state of each QS receptor controls biofilm formation and dispersal. Despite the inactivity of our exogenously supplied synthetic CAI-1 in biofilms, our experiments (Fig 5) demonstrate that endogenously produced CAI-1 is active within biofilms.

In contrast to what we find here, in which exogenous AI-2 is the strongest QS signal, previous studies, including from us, have reported that CAI-1 is the stronger of the two autoinducers in promoting the *V. cholerae* HCD QS mode [11,12,40]. These earlier conclusions were based on data from ΔcqsA and ΔluxS mutants that produce no CAI-1 or no AI-2, respectively. We now know that positive feedback on *cqsS* transcription occurs at HCD, while there is no evidence for feedback on *luxPQ* [40]. Indeed, the left panel of Fig 5B shows the cell-density–dependent increase that occurs in CqsS-3×FLAG relative to LuxQ-3×FLAG. This regulatory arrangement leads to increased CqsS levels relative to LuxPQ levels at HCD, abrogating the coincidence detection requirement. Apparently, QS coincidence detection is relevant only at cell densities below the threshold for activation of the CqsS positive feedback loop (feedback occurs roughly at $OD_{600} > 1$). Perhaps, once the cell-density condition is reached for positive feedback on *cqsS*, *V. cholerae* is at sufficiently high cell numbers that it commits to the planktonic lifestyle irrespective of the level of AI-2 in the vicinal community.

Here, we focused exclusively on *V. cholerae* El Tor biotype strain C6706, which possesses a functional QS system. Some *V. cholerae* strains of both the El Tor and Classical biotypes harbor mutations in *hapR* that render the HapR proteins nonfunctional [47]. Thus, in these strains, HapR-directed QS control of biofilm behavior does not occur. Notably, an alternative signal transduction system, called VieSAB, can also be involved in controlling biofilm phenotypes via modulation of the levels of the second messenger cyclic diguanylate, particularly in the Classical biotype [48–50]. In *V. cholerae* strains possessing functional HapR, *vieSAB* is repressed by HapR at HCD, so we propose that in these strains, QS controls biofilm dispersal. We speculate that in *V. cholerae* strains lacking a functional HapR, biofilm dispersal could be controlled by the VieSAB pathway. Moreover, given the reduced propensity for *V. cholerae hapR* mutants to disperse from biofilms (Fig 2D), we further speculate that such mutants primarily occupy niches in which it is advantageous for *V. cholerae* to remain in the biofilm state for long periods, possibly including under HCD conditions.

Collectively, this work, for the first time, to our knowledge, reveals the constraints enabling kin and nonkin QS signaling to occur in *V. cholerae*. Although both QS autoinducers work in

concert, *V. cholerae* relies on a census of the total bacteria in the local community, as measured by AI-2 concentration, to inform its decision to disperse from biofilms. For AI-2 to properly function as an interspecies signal, it is critical that kin community members do not saturate their AI-2 receptors with endogenously produced AI-2. Our work shows that *V. cholerae* avoids this circumstance by having a low cell-density threshold for activation by the kin CAI-1 molecule and a dramatically higher cell-density threshold for activation by the broadly made AI-2 molecule. We predict that other bacterial species that release and detect nonkin signals must employ analogous mechanisms to prevent tripping of their QS circuits absent an accurate estimation of the total cell density of the environment.

## Materials and methods

### Bacterial strains and reagents

The parent *V. cholerae* strain used in this study was WT O1 El Tor biotype C6706str2 [51]. Antibiotics, when necessary, were used at the following concentrations: ampicillin, 100 μg/mL; kanamycin, 50 μg/mL; polymyxin B, 50 μg/mL; streptomycin, 500 μg/mL; spectinomycin, 200 μg/mL; and chloramphenicol, 1 μg/mL. Strains were propagated in lysogeny broth (LB) supplemented with 1.5% agar or in liquid LB with shaking at 30°C. All strains used in this work are reported in S1 Table.

### DNA manipulation and strain construction

To generate DNA fragments used in natural transformations, including fusions and exchanges of *luxPQ* and *cqsS* alleles, splicing overlap extension (SOE) PCR was performed using iProof polymerase (Bio-Rad, Hercules, CA, USA) to combine DNA pieces. Primers used in this study are reported in S2 Table. In all cases, approximately 3 kb of upstream and downstream flanking regions generated by PCR from *V. cholerae* genomic DNA were included to ensure high chromosomal integration frequency. DNA fragments that were not native to *V. cholerae* were synthesized as g-blocks (IDT, Coralville, IA, USA) or were purchased as plasmids (mNG was licensed from Allele Biotech; San Diego, CA, USA) [52,53]. HapR was fused to mNG as previously described [10].

All *V. cholerae* strains constructed in this work were generated by replacing genomic DNA with DNA introduced by natural transformation (MuGENT) as recently described [54,55]. Briefly, the parent strain was grown overnight from a single colony at 30°C in liquid LB medium with agitation. The overnight culture was diluted 1:1,000 into fresh medium, and the strain was grown to an $OD_{600}$ of approximately 1.0. Cells were pelleted at 13,000 rpm in a microcentrifuge for 1 min and were resuspended at the original volume in 1× Instant Ocean (IO) Sea Salts (7 g/L) (Blacksburg, VA, USA). A 100 μL aliquot of this cell suspension was added to 900 μL of a chitin (Alfa Aesar, Haverhill, MA, USA) IO mixture (8 g/L chitin) and incubated overnight without agitation at 30°C. The next day, the DNA fragment containing the desired chromosomal alteration and an antibiotic resistance cassette for integration at the neutral locus *vc1807* were added to the cell-chitin preparation. This mixture was incubated for 12–24 h at 30°C without shaking, after which, excess IO was removed and replaced with liquid LB. The sample was vigorously shaken to remove *V. cholerae* cells from the chitin particles, and the preparation was dispensed onto LB plates containing relevant antibiotics followed by incubation at 30°C overnight. Resulting colonies were restreaked three times on LB plates with appropriate antibiotics, after which PCR and sequencing were used to verify correct integration of the introduced DNA fragments. Genomic DNA from these recombinant strains was used as a template for PCR to generate DNA fragments for future co-transformation, when necessary. Antibiotic resistance cassettes linked to Δ*vc1807* were a gift from Ankur Dalia.

## Real-time biofilm development and dispersal assay

Single *V. cholerae* colonies were grown overnight in a 96-well plate in 200 μL of LB medium with shaking at 30˚C covered with a breathe-easy membrane (Diversified Biotech, Dedham, MA, USA). The cultures were diluted 1:200 into fresh LB and subsequently grown for 7 h at 30˚C to an $OD_{600}$ of approximately 2.0. The cultures were diluted to an $OD_{600}$ of $1 \times 10^{-5}$, a roughly a 1:200,000 dilution in M9 medium containing glucose and casamino acids (1× M9 salts, 100 μM $CaCl_2$, 2 mM $MgSO_4$, 0.5% dextrose, 0.5% casamino acids). These cultures were dispensed onto No. 1.5 glass coverslip bottomed 96-well plates (MatTek, Ashland, MA, USA) and cells were allowed to attach for 1 h at 30˚C. Wells were washed to remove unattached cells by removing 200 μL of medium with a multichannel pipette and replacing with 200 μL of fresh M9 medium. After three washes, 200 μL of M9 medium was added to each well, and cultures were placed in a temperature-controlled chamber for microscopy (OKOlab, Ambridge, PA, USA) at 30˚C. Image acquisition was initiated 1 h later.

## Exogenous administration of synthetic autoinducers and agonists

Chemical syntheses of CAI-1, the AI-2 precursor, 4,5-dihydroxy-2,3-pentanedione (DPD), and the CqsS agonist Mimic$^{CAI-1}$ have been previously described [12,40,56,57]. Each compound was added to medium at a final concentration of 5 μM resulting in a final DMSO concentration of 0.25%. Control cultures were supplemented with 0.25% DMSO. For experiments involving AI-2, the medium was supplemented with 0.1 mM boric acid. In all cases, autoinducers were added to cells following attachment to the glass coverslips, i.e., immediately after the final washing step described above.

## Microscopy and image analysis

Imaging of growing and dispersing biofilms was performed using a DMI8 Leica SP-8 point scanning confocal microscope (Leica, Wetzlar, Germany). The light source for both fluorescence and brightfield microscopy was a tunable white-light laser (Leica; model #WLL2; excitation window = 470–670 nm). Biofilms were imaged using a 10× air objective (Leica, HC PL FLUOTAR; NA: 0.30) or a 63× water immersion objective (Leica, HC PL APO CS2; NA: 1.20) as indicated. For both transmission brightfield and confocal fluorescence microscopy, many wells in each plate were imaged simultaneously as specified in the Leica LasX software with a time interval of 30 min. The focal plane was maintained with adaptive focus control. A depth of 40 μm was sectioned with Nyquist sampling in XY and Z at each time point. Brightfield images were acquired at 640 nm, and light was detected in the transmitted path using a brightfield PMT for the Leica DMI stand. For fluorescence microscopy, excitation wavelengths of 503 and 558 nm were used for mNG and mRuby3, respectively. Sequential line scanning was performed to minimize spectral bleed-through in images. Emitted light was detected using GaAsP spectral detectors (Leica, HyD SP), and timed gate detection was employed to minimize the background signal.

Image analysis was performed in FIJI software (Version 1.52e). Biofilms were segmented in the brightfield images using an intensity threshold after image smoothing. The same threshold was applied to all images in this study. The total amount of light attenuated within each segmented area was summed for the entire imaging field at each time point, akin to a local optical density measurement. Data were exported for quantitation and graphing in R software using gglplot2 (https://ggplot2.tidyverse.org). In all plots, data were normalized to the reference strain/conditions for that day rather than as absolute biofilm biomass values because of slight variability in the amount of biofilm formation (across all strains) that occurred from day to day. In the case of fluorescence images, biofilms were initially segmented using the brightfield

approach described above. Total fluorescence signal from mNG and mRuby3 was subsequently measured from single biofilms and plotted in ggplot2.

## Bioluminescence assay

Three colonies of each strain to be analyzed were individually grown overnight in 200 μL LB with shaking at 30°C in a 96-well plate covered with a breathe-easy membrane. The following morning, the cultures were diluted 1:5,000 into fresh SOC medium or SOC medium containing the indicated concentrations of autoinducers. The plates were placed in a BioTek Synergy Neo2 Multi-Mode reader (BioTek, Winooski, VT, USA) with constant shaking at 30°C. Both $OD_{600}$ and bioluminescence from the chromosomally integrated *lux* operon were measured. Results were exported to R, and bioluminescence values were divided by $OD_{600}$ to produce RLUs. Results from the triplicate experiments were averaged and plotted using the ggplot2 plugin for R.

## Virulence factor production assay

To monitor virulence factor production, *V. cholerae* strains containing a chromosomal *tcpA-3×FLAG* fusion were grown from single colonies in liquid LB medium for 16 h. Cultures were diluted 1:5,000 into fresh AKI medium [58]. The cultures were incubated at 37°C without shaking for 4 h, followed by vigorous shaking for 2 h at 37°C. The cells were subjected to centrifugation for 2 min at 13,000 rpm, and the resulting pellets were flash frozen. Pellets were subsequently thawed, resuspended in 1× SDS-PAGE buffer, and boiled for 10 min at 95°C in preparation for SDS-PAGE and western blotting as described in the next section.

## Western blotting

Cultures of strains carrying CqsS-3×FLAG and LuxQ-3×FLAG were collected at the indicated $OD_{600}$ and subjected to centrifugation for 2 min at 13,000 rpm. The pellets were flash frozen, thawed, and lysed for 10 min at 25°C by resuspending in 75 μL Bug Buster (EMD Millipore, #70584–4; Danvers, MA, USA) supplemented with 0.5% Triton-X, 50 μL/mL lysozyme, 25 U/mL benzonase nuclease, and 1 mM phenylmethylsulfonyl fluoride (PMSF) per 1.0 $OD_{600}$ of pelleted culture. The cell lysate was solubilized in 1× SDS-PAGE buffer for 1 h at 37°C.

Samples with CqsS-3×FLAG, LuxQ-3×FLAG, or TcpA3X-FLAG were loaded onto 4%–20% Mini-Protein TGX gels (Bio-Rad). Electrophoresis was carried out at 200 V until the loading buffer reached the bottom of the gel. Proteins were transferred from the gels to PVDF membranes (Bio-Rad) for 1 h at 4°C at 100 V in 25 mM Tris buffer, 190 mM glycine, 20% methanol. Membranes were blocked for 1 h in PBST (137 mM NaCl, 2.7 mM KCl, 8 mM $Na_2HPO_4$, 2 mM $KH_2PO_4$, and 0.1% Tween) with 5% milk, followed by three washes with PBST. Subsequently, membranes were incubated for 1 h with a monoclonal anti-FLAG-peroxidase antibody (Millipore Sigma, #A8592; Danvers, MA, USA) at a 1:5,000 dilution in PBST with 5% milk. After washing four times with PBST for 10 min each, membranes were exposed using the Amersham ECL western blotting detection reagent (GE Healthcare, Chicago, IL, USA). For the RpoA loading control, the same protocol was followed except that the primary antibody was anti-*Escherichia coli* RNA polymerase α (Biolegend, #663104; San Diego, CA, USA) used at a 1:10,000 dilution and the secondary antibody was an anti-mouse IgG HRP conjugate antibody (Promega, #W4021; Madison, WI, USA) also used at a 1:10,000 dilution.

## Flow cytometry analyses

The secrete-and-sense strains constitutively produced mRuby3 enabling differentiation from the nonfluorescent sense-only strains. In all cases, strains were grown overnight with shaking

at 30°C, either in monoculture or as 1:1 cocultures of secrete-and-sense and sense-only strains. The cultures were diluted to $OD_{600}$ of $5 \times 10^{-6}$ at roughly a 1:500,000 dilution. Starting 2 h postinoculation, aliquots of cells were collected in 1-h intervals, and fixation was performed per safety protocol for performing flow cytometry with a BSL2 organism. Cells were pelleted in a microcentrifuge at 13,000 rpm for 1 min, resuspended in 100 μL of 3.7% formaldehyde (Electron Microscopy Sciences, Hatfield, PA, USA) in filter-sterilized PBS, and left at room temperature for 10 min. Subsequently, three washes were performed to remove excess formaldehyde. In the three washing steps, the cells were pelleted in a microcentrifuge at 13,000 rpm for 1 min and resuspended in 1 mL of PBS. After the final wash, cells were resuspended in 1 mL of PBS, except for LCD cultures, which were resuspended at 5× concentration in 200 μL PBS to increase the frequency of detection events in the subsequent flow cytometry analysis. Following fixation and washing, cells were stored at 4°C in the dark until flow cytometry was performed. mRuby3 and mNG fluorescence signals were compared before and after fixation by microscopy, and no fluorescence signal was lost during fixation.

Flow cytometry was performed on samples using a FACSAria Special Order Research Product driven by FACSDiva software (BD Biosciences, San Jose, CA, USA). A 561-nm laser line was used to excite mRuby3 fluorescence, and a 488-nm laser line was used for mNG fluorescence. Forward and side scatter were used to gate a distinct single cell population, and within this gate, two distinct peaks were identified in the mRuby3 channel corresponding to cells that strongly produced the mRuby3 fluorescent protein (secrete-and-sense cells) and those that did not (sense-only cells). Cells were further gated based on this histogram to assign appropriate mNG signals to the secrete-and-sense and sense-only cell populations. Data from all samples were collected with identical gates, laser intensity, and PMT voltages.

## Supporting information

**S1 Fig. Complementation of Δ*vpsL* and Δ*hapR* mutant phenotypes.** (A) Quantitation of biofilm biomass for *V. cholerae* WT, the Δ*vpsL* strain, and the complemented Δ*vpsL lacZ*::P*vpsL*-*vpsL* strain over time. (B) As in A for *V. cholerae* WT, the Δ*hapR* strain, and the complemented Δ*hapR lacZ*::P*hapR*-*hapR* strain. Data are represented as means normalized to the peak biofilm biomass of the WT strain in each experiment. In all cases, $n = 3$ biological and $n = 3$ technical replicates, ± SD (shaded). Numerical data are available in S1 Data. WT, wild type. (TIF)

**S2 Fig. Response of the *V. cholerae* CAI-1 reporter strain to exogenous CAI-1 and Mimic^CAI-1.** (A) Left panel: representative projections of the *V. cholerae* CAI-1 reporter strain (Δ*vpsS*, Δ*cqsR*, Δ*luxQ*, Δ*cqsA*) treated with 0.25% DMSO (Ctrl), 5 μM CAI-1, or 5 μM Mimic^CAI-1 after 9 h of biofilm growth at 30°C. Right panel: quantitation of biofilm biomass for the strain in A treated with 0.25% DMSO (Ctrl), 5 μM CAI-1, or 5 μM Mimic^CAI-1 over time. Data are represented as means normalized to the peak biofilm biomass of the DMSO control strain. $n = 3$ biological and $n = 3$ technical replicates, ± SD (shaded). (B) The corresponding *lux* pattern for the strain in A following treatment with 0.25% DMSO (Ctrl) or 5 μM CAI-1. (C) As in B following treatment with 0.25% DMSO (Ctrl) or 5 μM Mimic^CAI-1. RLUs are defined as light production (a.u.) divided by $OD_{600}$. For B and C, $n = 3$ biological replicates and error bars represent SD. Numerical data are available in S1 Data. a.u., arbitrary unit; CAI-1, *cholerae* autoinducer-1; Ctrl, control; OD, optical density; RLU, relative light unit. (TIF)

**S3 Fig. Response of the *V. cholerae* AI-2 reporter strain to exogenous AI-2.** (A) Left panel: representative projections of the *V. cholerae* AI-2 reporter strain (Δ*vpsS*, Δ*cqsR*, Δ*cqsS*, Δ*luxS*)

treated with 0.25% DMSO (Ctrl) or 5 μM AI-2 after 9 h of biofilm growth at 30°C. Right panel: Quantitation of biofilm biomass for the strain in A treated with 0.25% DMSO (Ctrl) or 5 μM AI-2 over time. Data are represented as means normalized to the peak biofilm biomass of the DMSO control strain. $n = 3$ biological and $n = 3$ technical replicates, ± SD (shaded). (B) The corresponding *lux* pattern for the strain in A following treatment with 0.25% DMSO (Ctrl) or 5 μM AI-2. RLUs are defined as light production (a.u.) divided by $OD_{600}$. n = 3 biological replicates and error bars represent SD. Numerical data are available in S1 Data. AI-2, autoinducer-2; a.u., arbitrary unit; Ctrl, control; OD, optical density; RLU, relative light unit. (TIF)

**S4 Fig. Exogenous AI-2 represses biofilm formation in the Δ*vpsS*, Δ*cqsR* *V. cholerae* strain, but Mimic$^{CAI-1}$ does not.** Quantitation of biofilm biomass for the *V. cholerae* Δ*vpsS*, Δ*cqsR* strain treated with 0.25% DMSO (Ctrl), 5 μM Mimic$^{CAI-1}$, or 5 μM AI-2 over time. Data are represented as means normalized to the peak biofilm biomass of the DMSO control strain in each experiment. $n = 3$ biological and $n = 3$ technical replicates, ± SD (shaded). Numerical data are available in S1 Data. AI-2, autoinducer-2; Ctrl, control. (TIF)

**S5 Fig. Exogenous AI-2 activates WT *V. cholerae* *lux* expression, but Mimic$^{CAI-1}$ does not.** (A) The *lux* pattern for WT *V. cholerae* over time. (B) As in A following treatment with 0.25% DMSO (Ctrl), 5 μM Mimic$^{CAI-1}$, or 5 μM AI-2. (C) As in B for the Δ*vpsS*, Δ*cqsR* strain. RLUs are defined as light production (a.u.) divided by $OD_{600}$. $n = 3$ biological replicates and error bars represent SD. Numerical data are available in S1 Data. AI-2, autoinducer-2; a.u., arbitrary unit; CAI-1, *cholerae* autoinducer-1; Ctrl, control; OD, optical density; RLU, relative light unit; WT, wild type. (TIF)

**S6 Fig. LuxPQ, but not CqsS, drives virulence factor production at LCD.** Representative western blot showing TcpA-3×FLAG in the *V. cholerae* strain possessing both the CqsS and LuxPQ QS circuits (AI-2$^{S+R+}$, CAI-1$^{S+R+}$; first lane), lacking all QS receptors (AI-2$^{S+R-}$, CAI-1$^{S+R-}$; second lane), possessing only the CAI-1 QS circuit (CAI-1$^{S+R+}$; third lane), and possessing only the AI-2 QS circuit (AI-2$^{S+R+}$, fourth lane). RpoA was used as the loading control. Quantification is based on three biological replicates for each condition. Values were normalized to the strain possessing both QS circuits. Numerical data are available in S1 Data. AI-2, autoinducer-2; CAI-1, *cholerae* autoinducer-1; LCD, low cell density; QS, quorum sensing. (TIF)

**S7 Fig. Single synthase mutants display biofilm dispersal defects.** Left panel: schematic representing *V. cholerae* strains used in the right panel. Right panel: quantitation of biofilm biomass over time for the strain possessing both QS receptors and synthases (AI-2$^{S+R+}$, CAI-1$^{S+R+}$; blue), both QS receptors but lacking *luxS* (AI-2$^{S-R+}$, CAI-1$^{S+R+}$; green), and both QS receptors but lacking *cqsA* (AI-2$^{S+R+}$, CAI-1$^{S-R+}$; red). Data are represented as means normalized to the peak biofilm biomass of the WT strain in each experiment. In all cases, $n = 3$ biological and $n = 3$ technical replicates, ± SD (shaded). Numerical data are available in S1 Data. AI-2, autoinducer-2; QS, quorum sensing; WT, wild type. (TIF)

**S1 Movie. Time-lapse video of *V. cholerae* biofilm lifecycle for the indicated strains as imaged by brightfield microscopy.** (AVI)

**S2 Movie. Time-lapse video of AphA-mNG or HapR-mNG during the biofilm lifecycle of otherwise WT *V. cholerae*.** mNG, mNeonGreen; WT, wild type.
(AVI)

**S1 Table. List of strains used in this study.**
(DOCX)

**S2 Table. List of oligonucleotides used in this study.**
(DOCX)

**S1 Data. Numerical data for Figs 2, 3, 4, 5, 6 and 7 and S1, S2, S3, S4, S5 and S7.**
(XLSX)

**S1 Raw Images. Raw gel images for Figs 4 and 5 and S6.**
(PDF)

## Acknowledgments

We thank members of the Bassler group and Prof. Ned Wingreen for thoughtful discussions. We particularly thank Dr. Ameya Mashruwala for providing the luciferase reporter used in this study.

## Author Contributions

**Conceptualization:** Andrew A. Bridges, Bonnie L. Bassler.

**Data curation:** Andrew A. Bridges, Bonnie L. Bassler.

**Formal analysis:** Andrew A. Bridges, Bonnie L. Bassler.

**Funding acquisition:** Bonnie L. Bassler.

**Investigation:** Andrew A. Bridges, Bonnie L. Bassler.

**Methodology:** Andrew A. Bridges.

**Project administration:** Bonnie L. Bassler.

**Resources:** Andrew A. Bridges, Bonnie L. Bassler.

**Supervision:** Bonnie L. Bassler.

**Validation:** Andrew A. Bridges, Bonnie L. Bassler.

**Visualization:** Andrew A. Bridges, Bonnie L. Bassler.

**Writing – original draft:** Andrew A. Bridges, Bonnie L. Bassler.

**Writing – review & editing:** Andrew A. Bridges, Bonnie L. Bassler.

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
