## [Editor Report · Decision Letter 0]

18 Jul 2019

Dear Dr Bassler, 

Thank you for submitting your manuscript entitled "The intra-genus and inter-species quorum-sensing autoinducers exert distinct control over Vibrio cholerae biofilm formation and dispersal" for consideration as a Research Article by PLOS Biology.

Your manuscript has now been evaluated by the PLOS Biology editorial staff, as well as by an academic editor with relevant expertise, and I'm writing to let you know that we would like to send your submission out for external peer review.

**Important**: Please also see below for further information regarding completing the MDAR reporting checklist. The checklist can be accessed here: https://plos.io/MDARChecklist

Please re-submit your manuscript and the checklist, within two working days, i.e. by Jul 22 2019 11:59PM.

Kind regards,

Roli Roberts

Senior Editor

PLOS Biology

INFORMATION REGARDING THE REPORTING CHECKLIST:

PLOS Biology is pleased to support the "minimum reporting standards in the life sciences" initiative (https://osf.io/preprints/metaarxiv/9sm4x/). This effort brings together a number of leading journals and reproducibility experts to develop minimum expectations for reporting information about Materials (including data and code), Design, Analysis and Reporting (MDAR) in published papers. We believe broad alignment on these standards will be to the benefit of authors, reviewers, journals and the wider research community and will help drive better practise in publishing reproducible research. 

We are therefore participating in a community pilot involving a small number of life science journals to test the MDAR checklist. The checklist is intended to help authors, reviewers and editors adopt and implement the minimum reporting framework. 

IMPORTANT: We have chosen your manuscript to participate in this trial. The relevant documents can be located here:

MDAR reporting checklist (to be filled in by you): https://plos.io/MDARChecklist

**We strongly encourage you to complete the MDAR reporting checklist and return it to us with your full submission, as described above. We would also be very grateful if you could complete this author survey:

https://forms.gle/seEgCrDtM6GLKFGQA

Additional background information:

Interpreting the MDAR Framework: https://plos.io/MDARFramework

Please note that your completed checklist and survey will be shared with the minimum reporting standards working group. However, the working group will not be provided with access to the manuscript or any other confidential information including author identities, manuscript titles or abstracts. Feedback from this process will be used to consider next steps, which might include revisions to the content of the checklist. Data and materials from this initial trial will be publicly shared in September 2019. Data will only be provided in aggregate form and will not be parsed by individual article or by journal, so as to respect the confidentiality of responses. 

Please treat the checklist and elaboration as confidential as public release is planned for September 2019.

We would be grateful for any feedback you may have.

---

## [Decision Letter · Decision Letter 1]

28 Aug 2019

Dear Dr Bassler,

Thank you very much for submitting your manuscript "The intra-genus and inter-species quorum-sensing autoinducers exert distinct control over Vibrio cholerae biofilm formation and dispersal" for consideration as a Research Article at PLOS Biology. Your manuscript has been evaluated by the PLOS Biology editors, an Academic Editor with relevant expertise, and by four independent reviewers.

You'll see that all four of the reviewers are broadly positive about the manuscript; however, each raises a number of concerns that must be addressed before further consideration. In light of the reviews (below), we will not be able to accept the current version of the manuscript, but we would welcome resubmission of a much-revised version that takes into account the reviewers' comments. We cannot make any decision about publication until we have seen the revised manuscript and your response to the reviewers' comments. Your revised manuscript is also likely to be sent for further evaluation by the reviewers.

Your revisions should address the specific points made by each reviewer. Please submit a file detailing your responses to the editorial requests and a point-by-point response to all of the reviewers' comments that indicates the changes you have made to the manuscript. In addition to a clean copy of the manuscript, please upload a 'track-changes' version of your manuscript that specifies the edits made. This should be uploaded as a "Related" file type. You should also cite any additional relevant literature that has been published since the original submission and mention any additional citations in your response. 

Before you revise your manuscript, please review the following PLOS policy and formatting requirements checklist PDF: http://journals.plos.org/plosbiology/s/file?id=9411/plos-biology-formatting-checklist.pdf. It is helpful if you format your revision according to our requirements - should your paper subsequently be accepted, this will save time at the acceptance stage.

Please note that as a condition of publication PLOS' data policy (http://journals.plos.org/plosbiology/s/data-availability) requires that you make available all data used to draw the conclusions arrived at in your manuscript. If you have not already done so, you must include any data used in your manuscript either in appropriate repositories, within the body of the manuscript, or as supporting information (N.B. this includes any numerical values that were used to generate graphs, histograms etc.). For an example see here: http://www.plosbiology.org/article/info%3Adoi%2F10.1371%2Fjournal.pbio.1001908#s5.

For manuscripts submitted on or after 1st July 2019, we require the original, uncropped and minimally adjusted images supporting all blot and gel results reported in an article's figures or Supporting Information files. We will require these files before a manuscript can be accepted so please prepare them now, if you have not already uploaded them. Please carefully read our guidelines for how to prepare and upload this data: https://journals.plos.org/plosbiology/s/figures#loc-blot-and-gel-reporting-requirements.

Upon resubmission, the editors will assess your revision and if the editors and Academic Editor feel that the revised manuscript remains appropriate for the journal, we will send the manuscript for re-review. We aim to consult the same Academic Editor and reviewers for revised manuscripts but may consult others if needed.

We expect to receive your revised manuscript within two months. Please email us (plosbiology@plos.org) to discuss this if you have any questions or concerns, or would like to request an extension. At this stage, your manuscript remains formally under active consideration at our journal; please notify us by email if you do not wish to submit a revision and instead wish to pursue publication elsewhere, so that we may end consideration of the manuscript at PLOS Biology.

When you are ready to submit a revised version of your manuscript, please go to https://www.editorialmanager.com/pbiology/ and log in as an Author. Click the link labelled 'Submissions Needing Revision' where you will find your submission record. 

Sincerely,

Roli Roberts

Senior Editor

PLOS Biology

REVIEWERS' COMMENTS:

Reviewer #1:

The manuscript describes the interaction of two QS systems, CAI and AI-2 in V. cholerae to better define how these two systems can impinge on the same phosphorelay system, LuxO/U, to control phenotypic responses. The work largely focuses on a number of genetic constructs to determine these interactions. The authors conclude that CAI-1 is used to detect self and AI-2 to detect non-self and that the QS is largely driven by the AI-2 system. 

The authors have clearly undertaken a considerable amount of work here to tease out these effects. One consequence is that the writing and text is very dense and hence it is hard to follow at times. There is a fair bit of narrative in the results, which if removed might help with the focus on interpreting the outcomes. Some of these are quite subjective, e.g. impervious, exquisite etc and are prone to hyperbole. While it is appreciated that the authors used a different biofilm set up, this section could be omitted or considerably shortened to cut down on the text. Similarly, given that the-self signalling experiments are convoluted and suggest there is no self signalling, that data can be omitted—although the final conclusions suggest that V. cholerae uses these as self sensing systems. 

The authors argue throughout the manuscript that it is a density dependent phenomenon observed. However, the QS field seems to largely acknowledge that the accumulation of signals due to local signal concentration may also regulate these phenotypes. Since their system, and plankton cells are in a closed system, the signal concentration will track with cell number and hence it is not possible to differentiate the two. Additionally, batch systems will become limited in oxygen and nutrients due to metabolism, which also tracks with cell number and thus, it is thus possible that the batch biofilm system gives different results for any of these reasons. Given that they see the same changes in AphA and HapR in both planktonic and biofilm cells, it suggests this is not a biofilm specific phenotype. 

One of the biggest concerns is that the chemically synthesised CAI-1 does not induce the CAI-1 pathway for biofilms. In contrast a completely different chemical does. Thus, how can the authors justify that this system is responding to CAI-1? This has significant consequences for the rest of the data dependent on adding back the alternative chemical. It would also be important to not call the artificial compound CAI-1* as that is easily confused with the native signal CAI-1. A clearly different name should be used to help the reader. 

Why would CAI-1 form micelles for the biofilm culture but not the planktonic culture at the same concentrations? 

The Western blot data should be quantified as the differences are quite subtle for some. For example, it is difficult to judge from the image if CqsS is more abundant that LuxQ at high density. How do the authors rationalise that CqsS is really only active at LCD, but that the protein is considerably more highly expressed at HCD, while there is only an apparently modest increase in LuxQ, despite it essentially being the dominant system here. 

Why were experiments conducted at 30C in M9 salts medium, which has low NaCl compared to most work conducted with Vibrio species and has a complex nutrient and nitrogen source of glucose and casamino acids,? Perhaps this goes a long way to explaining the differences obtained here compared to the other studies, as noted in the discussion. 

The AI-2 is resented as a ‘universal’ signal and this does not seem to really be the case. For the Vibrio spp, E. coli and Salmonella, this seems correct, but in many other species, it has been shown to have no true signalling function, or at least none that have yet been determined. Similarly, biofilm formation is QS controlled for some, but not all bacteria and these points should be amended for accuracy. 

Were the signals added from time 0 of the experiments? 

One suggestion might be that the integration of CAI-1 and AI-2 is one of biochemistry, dependent on signal-receptor binding constants, receptor numbers, as well as kinase/phosphatase activity rates. thus, it may be cleaner to compare these for purified components if possible given all of the complexities of the various interacting regulators, differences in biofilm/planktonic responses for the artificial vs real CAI signal etc. Clearly, not in the scope of this manuscript though.

Reviewer #2:

The study by Bridges and Bassler focuses on the role of quorum sensing (QS) for biofilm formation and dispersal of the bacterial pathogen Vibrio cholerae. Specifically, the authors analyze two autoinducer molecules, CAI-1 and AI-2, which signal through distinct membrane-bound receptors (sensors) to the same response regulator, LuxO. To this end, the authors establish a live-cell biofilm assay, and show that the QS system functions as a co-incidence detector, since both autoinducers need to be present simultaneously to trigger phosphate flow reversion (sensor kinase to phosphatase) and repression of biofilm formation. Intriguingly, CAI-1-signaling indeed occurs by QS not by “self sensing”, and the system responds with much greater sensitivity to CAI-1 than to AI-2, suggesting that the detection of kin is a prerequisite for the detection of and response to non-kin bacteria.

In summary, this interesting and thorough study addresses and significantly contributes to answering the longstanding question of how different signaling pathways, which coincide on a common response regulator, contribute to control complex bacterial behavior like biofilm formation or virulence. Based on the concepts of coincidence detection and vastly different thresholds at which CAI-1 and AI-2 trigger signaling, it appears that V. cholerae engages on biofilm dispersal when an area gets “overcrowded” by non-kin bacteria and only forms a new biofilm in an unoccupied area.

The study is well controlled, and the story unravels in a straightforward manner. A few rather minor points should be addressed to further improve the manuscript.

Specific points

1) Fig. 2BD: The impressive biofilm phenotypes of the V. cholerae mutant strains lacking vpsL or hapR should be complemented by providing the corresponding genes on a plasmid.

2) Fig. 3BD: For the mRuby3 fluorescence, standard deviation (shading) is shown but not the mean data points. The corresponding data points and lines should be added.

3) Fig. 5E: The “green circles” are actually black circles on a green line. Please reconcile.

4) Fig. S1A: Bright field projections of V. cholerae biofilms (as in Fig. 4A) would be informative.

5) Unexpectedly, synthetic CAI-1 is inactive under the live-cell biofilm formation conditions used in the study, while active under planktonic conditions in shaken culture. The possible reason given by the authors – “micelle formation” (l.203) – does not seem to be overtly plausible, since in both cases the same medium was used (and hence, micelle formation of CAI-1 is likely similar). Also, the results are surprising in light of the finding that CAI-1-mediated signaling is indeed QS not “self-sensing” (Fig. 7). These observations should be more thoroughly outlined in the Discussion section.

6) The authors address a possible dose-response relationship of sensor kinase (CqsS, LuxQ) production and QS in the course of biofilm formation and dispersal (Fig. 5). The observations could also be discussed in the context of Vibrio QS autoinducers acting dose-dependently as “timers” – at least under planktonic conditions (Anetzberger et al., 2012, PLoS One 10: e48310).

7) l.306: … was similar in the … (not “identical”)

Reviewer #3:

Vibrio cholerae controls major processes such as biofilm formation and virulence via cell-cell communication. Several systems are involved in this regulation, in particular the CAI-1-dependent and AI-2-dependent systems. CAI-1 and AI-2 are autoinducers used by V. cholerae to measure its own abundance and the abundance of total bacterial cells, respectively. The two autoinducers bind distinct receptors but the signal is relayed to the final effectors through the same pathway. This layout has made it difficult to understand the benefits of each part of the system. 

The authors sought to examine the contribution of each autoinducer to the transmission of the information and thus to the control of V. cholerae behavior. For that purpose, they developed a live-cell biofilm formation and dispersal assay.

This paper asks a very interesting question of great importance to understand the biology of V. cholerae and its behavior during its life-cycle. 

The paper presents elegant experiments, is well-written, well organized and easy to read. 

However, I have a few comments that are detailed below. 

MAJOR COMMENTS

1. Fig. 7 and lines 364 to 366: This is a nice experiment. However, its presentation is a bit too "black-and-white". The authors write that "if self sensing occurs, in co-culture, the autoinducer made by the secrete-and-sense strain would trigger its HCD mode, while the sense-only strain would remain in LCD mode". I think this statement (+ Fig. 7B) should be toned down by adding that another possibility would be that the sense-only cells could also display a delayed response compared to the secrete-and-sense cells. Also, is the difference that we see between the green and black lines in Fig. 7D not significant? Couldn't that reflect a difference in response between the 2 strains? If so, scenario 1 is to be put forward. 

Besides that, considering that the interest of the authors is to examine the role of the autoinducers in biofilm formation, it would be very informative to perform the same experiment in biofilm-forming conditions. This would factor in the fact that the autoinducers probably display a different availability than in shaken cultures. 

2. Great efforts have been made for the description of the experiments in the materials and methods section. However, I did not see any precise description of strain construction. It is necessary to add that information, preferentially to the supplementary material.

MINOR COMMENTS

1. Line 198: Please explain in one sentence how the heterologous readout for HapR-controlled QS activity works (which promoter is used for example). 

2. Lines 201 to 204: The authors state that synthetic CAI-1 is inactive in biofilm conditions probably because of micelle formation. Please briefly explain why (why would there be micelle formation and why these micelles would perturb the functioning of CAI-1 and not CAI-1*). Furthermore, on Fig. S1, although CAI-1 and CAI-1* both activate the production of light, the response profiles are different. Indeed, CAI-1* induces a very sharp response compared to CA1-1. Although this does probably not explain the disparity in biofilm formation, this difference in activity would be worth mentioning in the text. 

3. Please indicate in the materials and methods section (microscopy paragraph) why the biofilm data are normalized to the peak biofilm biomass for the control strain. 

4. Fig. 5E middle panel and line 308: Albeit in a delayed manner, the biofilm still disintegrates. Please specify what signal could induce the dispersal. 

5. Fig. 6: I wish the authors had also presented biofilm assays. Although lower than that of the wt, the patterns of light production are different in the red and green strain and it would have been interesting to see if this would translate to a different biofilm formation/dispersal behavior.

6. The authors perform their experiments in strain V. cholerae O1 El Tor which harbors the pathway leading to biofilm formation and virulence as described in Fig. 1. Other strains with a Classical background do not possess HapR. Could the authors discuss their results in light of this consideration?

Reviewer #4:

I have read “The intra-genus and inter-species quorum-sensing autoinducers exert distinct control over Vibrio cholerae biofilm formation and dispersal” by Bridges and Bassler. The paper increases our understanding of the wiring of the well-studied quorum sensing system of biofilm formation and dispersal in V. cholerae. The paper presents elegant real time microscopy methods to detect transition from biofilm growth to dispersal. The paper is well written, and the data are presented clearly and convincingly. I recommend publication but suggest the authors consider additional evolutionary interpretations of the CAI-1 and AI-2 coincidence detector mechanism.

The most intriguing finding is that the two studied QS pathways, CAI-1 and AI-2, differ so much with respect to the densities at which they switch from biofilm promoting to dispersal inducing. I am convinced by this difference. QS signals that are not unique to a genotype have been discussed at length in the evolutionary theory literature: how is honest signaling possible in face of potential signal cheats and what is it that QS is measuring? Here, the authors propose that the two signals act as a coincidence detector: only when both signals are present will dispersal from the biofilm be induced. The authors interpret this evolutionarily, too: when the inter-species signal AI-2 is present due to non-kins present in a habitat, a small nascent community of V. c. would rapidly disperse due to high sensitivity to the “private” CAI-1 signal; this is interpreted as a mechanism to disperse and find more suitable habitats. Additionally, I think this data---quite beautifully---corroborates theory that predicts QS may evolve robustly as a means to infer kinship. Because bacteria grow in clonal patches in biofilms, therefore, a brief waiting period “after landing” is sufficient to allow for the evolution of more complex cooperative phenotypes that positively feedback and increase the genotypes fitness (Schluter…Foster & Mitri, 2016 PLoS Comput Biol). The CAI-1/AI-2 pathway may be first empirical evidence for this logic, with AI-2 acting as an additional measure for crowdedness that is presumably inversely related to the residual capacity of the habitat. 

I recommend publication and append some minor comments that may improve the manuscript.

- The article is rather long. The section (line 385 ff) on self-sensing vs QS seems a bit unnecessary. If I understand correctly, these are results from shaken cultures. In biofilms, however, local kinship can be inferred due to clonal growth as relatedness increases naturally over time in such spatially structured situations. Here, therefore, the vastly different sensitivity thresholds to the two signals may hold the key to sensing self from non-self. 

- Figure 7B is poorly explained and the legend refers to the main text for details (of which there are not many). A better legend (e.g. what are the shadings of the cells?) would improve the figure.

- Figure 3: Was the AphA tagged experiment also started from an HCD overnight culture? Would we not expect, therefore, AphA to be low in the first hours, then high, and then decline (as is indeed the case for ahpa:hapr)?

- Can you comment on the observed or estimated diffusion rates of the two signals?

---

## [Decision Letter · Decision Letter 2]

14 Oct 2019

Dear Dr Bassler,

Thank you for submitting your revised Research Article entitled "The intra-genus and inter-species quorum-sensing autoinducers exert distinct control over Vibrio cholerae biofilm formation and dispersal" for publication in PLOS Biology. I've now obtained advice from two of the original reviewers and have discussed their comments with the Academic Editor. 

We're delighted to let you know that we're now editorially satisfied with your manuscript. However before we can formally accept your paper and consider it "in press", we also need to ensure that your article conforms to our guidelines. A member of our team will be in touch shortly with a set of requests. As we can't proceed until these requirements are met, your swift response will help prevent delays to publication.

IMPORTANT: Many thanks for supplying the supplementary data file. Please could you re-name it "S1_Data" and cite it in each of the relevant main and supplementary Figure legends (i.e. those for Figs 2, 3, 4, 5, 6, 7, S1, S2, S3, S4, S5, S7)?

Please note that you may have the opportunity to make the peer review history publicly available. The record will include editor decision letters (with reviews) and your responses to reviewer comments. If eligible, we will contact you to opt in or out.

Early Version: Please note that an uncorrected proof of your manuscript will be published online ahead of the final version, unless you opted out when submitting your manuscript. If, for any reason, you do not want an earlier version of your manuscript published online, uncheck the box. Should you, your institution's press office or the journal office choose to press release your paper, you will automatically be opted out of early publication. We ask that you notify us as soon as possible if you or your institution is planning to press release the article.

Sincerely,

Roli Roberts

Senior Editor

PLOS Biology

For manuscripts submitted on or after 1st July 2019, we require the original, uncropped and minimally adjusted images supporting all blot and gel results reported in an article's figures or Supporting Information files. We will require these files before a manuscript can be accepted so please prepare them now, if you have not already uploaded them. Please carefully read our guidelines for how to prepare and upload this data: https://journals.plos.org/plosbiology/s/figures#loc-blot-and-gel-reporting-requirements.

REVIEWERS' COMMENTS:

Reviewer #2:

[identifies himself as Hubert Hilbi]

The authors did a thorough and convincing job revising the manuscript and addressing the reviewers’ points in a satisfactory manner.

Reviewer #3:

The authors have answered the questions appropriately.

---

## [Editor Report · Decision Letter 3]

24 Oct 2019

Dear Dr Bassler,

On behalf of my colleagues and the Academic Editor, Victor Sourjik, I am pleased to inform you that we will be delighted to publish your Research Article in PLOS Biology. 

Early Version

PRESS 

Kind regards,

Hannah Harwood

Publication Assistant, 

PLOS Biology

on behalf of

Roland Roberts,

Senior Editor

PLOS Biology